# Broadband Spectral Induced Polarization for the detection of Permafrost and an approach to ice content estimation – A Case study from Yakutia, Russia

Jan Mudler[1], Andreas Hördt[1], Dennis Kreith[1], Madhuri Sugand[1], Kirill Bazhin[2], Lyudmila Lebedeva[2], and Tino Radić[3]

[1]Technische Universität Braunschweig, Institut für Geophysik und extraterrestrische Physik, Braunschweig, Germany
[2]Melnikov Permafrost Institute, Russian Academy of Science, Yakutsk, Russia
[3]Radic Research, Berlin, Germany
**Correspondence:** Jan Mudler (j.mudler@tu-bs.de)

**Abstract.** The reliable detection of subsurface ice using non-destructive geophysical methods is an important objective in permafrost research. The ice content of the frozen ground is an essential parameter for further interpretation, for example in terms of risk analysis and for the description of permafrost carbon feedback by thawing processes.

The High-Frequency Induced Polarization method (HFIP) enables the measurement of the frequency dependent electrical con-

ductivity and permittivity of the subsurface, in a frequency range between $100\,\mathrm{Hz}$ and $100\,\mathrm{kHz}$. As the electrical permittivity of ice exhibits a strong characteristic behaviour in this frequency range, HFIP in principle is suitable to estimate ice content. Here, we present methodological advancements of the HFIP method and suggest an explicit procedure for ice content estimation.

A new measuring device, the Chameleon-II (Radic Research), was used for the first time. Compared to a previous generation, the new system is equipped with longer cables and higher power, such that we can now achieve larger penetration depths up

to $10\,\mathrm{m}$. Moreover, it is equipped with technology to reduce electromagnetic coupling effects which can distort the desired subsurface signal.

The second development is a method to estimate ice content quantitatively from five Cole-Cole parameters obtained from spectral two-dimensional inversion results. The method is based on a description of the subsurface as a mixture of two components (matrix and ice) and uses a previously suggested relationship between frequency-dependent electrical permittivity and

ice content. In this model, the ice relaxation is considered the dominant process in the frequency range around $10\,\mathrm{kHz}$.

Measurements on a permafrost site near Yakutsk, Russia, were carried out to test the entire procedure under real conditions at the field scale. We demonstrate that the spectral signal of ice can clearly be identified even in the raw data, and show that the spectral 2-D inversion algorithm is suitable to obtain the multidimensional distribution of electrical parameters. The parameter distribution and the estimated ice content agree reasonably well with previous knowledge of the field site from borehole

and geophysical investigations. We conclude that the method is able to provide quantitative ice content estimates, and that relationships that have been tested in the laboratory may be applied at the field scale.

# 1 Introduction

The frequency dependent electrical properties of ice have been studied by several authors over the past decades in the laboratory for pure ice as well as for ice within sediment mixtures (e.g. Auty and Cole, 1952; Olhoeft, 1977; Hippel, 1988; Petrenko, 1993; Petrenko and Ryzhkin, 1997; Bittelli et al., 2004; Grimm et al., 2015; Artemov, 2019; Coperey et al., 2019). A limited number of field studies has been reported as well (e.g. Grimm and Stillman, 2015; Przyklenk et al., 2016). The permittivity of water ice exhibits a characteristic frequency-dependence in the frequency range between $100\,\mathrm{Hz}$ and $100\,\mathrm{kHz}$ (Petrenko and Whitworth, 2002; Artemov and Volkov, 2014). At the field scale, ice often does not occur in its pure form within the subsurface, but as mixtures with sediments, such as in frozen ground. The ice content of permafrost is an important piece of information, e.g. for risk analysis for humans and infrastructure (Hauck and Kneisel, 2008; Heginbottom et al., 2012) and for the description of the permafrost carbon feedback by thawing process due to climate change (Schuur et al., 2015).

In geophysics, models for the estimation of ice content use the contrast of characteristic physical material properties to distinguish ice from the other components of the subsoil. Since electrical resistivity tomography (ERT) does not provide enough information for a quantitative ice content estimation, Hauck et al. (2011) use a combination of ERT and seismic measurements to separate the subsoil into four components: the soil matrix (i.e. rock), ice, water and air. A joint 2-D inversion based on this model was presented by Wagner et al. (2019) and successfully applied for ice content estimation to several alpine field sites by Mollaret et al. (2020).

To estimate ice content with one method alone, the frequency-dependent electrical permittivity is a promising parameter. To describe electrical permittivity of ice-containing materials, models consisting of four and three components have been suggested by Bittelli et al. (2004) and Stillman and Grimm (2010). Theoretical investigations about the frequency-dependent electrical behaviour of frozen ground due to the polarization processes of ice were carried out by Kozhevnikov and Antonov (2012) and Zorin and Ageev (2017), with the aim to relate ice content directly to electrical permittivity. They assume fully saturated and frozen conditions, in which case the number of phases reduces to two (ice and matrix). An empirical approach was suggested by Grimm and Stillman (2015) who did not use the full spectral information but calculated the difference between electrical resistivity at two discrete frequencies. The calibration of this difference to estimate ice content is then based on laboratory investigations.

While the relationship between ice content and electrical permittivity has been well studied in the laboratory and with theoretical investigations, there is almost no experience at the field scale. A common geophysical method for the acquisition of the frequency dependent electrical signal of the subsurface is the Spectral Induced Polarization (SIP). While conventional SIP is typically used in the frequency range $< 1\,\mathrm{kHz}$ (Kemna et al., 2000), the measurement of the full process of ice relaxation requires the usage of high-frequency IP (HFIP) with an expanded frequency range up to $100\,\mathrm{kHz}$. The first application of broad-band detection of electrical parameters at the field scale was reported by Grimm and Stillman (2015). Przyklenk et al. (2016) obtained field data over pure ice with a system specifically designed for broadband data acquisition in combination with capacitive coupling. They discussed different parameterizations of frequency-dependent permittivity and the possibilities to resolve the parameters based on a homogeneous halfspace assumption. Mudler et al. (2019) discussed several case histo-

ries in different regions and suggested an approach for a two-dimensional inversion. Limbrock and Weigand (2020) obtained broadband data from alpine permafrost sites by field measurements and from laboratory investigations. They investigated the spectral signal in terms of thermal ground characteristics.

In the studies discussed in the previous paragraph, no attempts were made at a quantitative estimation of ice content based on full spectral information. This was partly because the focus was on different aspects, such as data inversion, capacitive coupling and feasibility of the equipment, as well as the thermal state. Moreover, the penetration depth of the previous generation of the Chameleon equipment was not sufficient to fully characterize the frozen ground in the relevant depth ranges. Here, we present the next logical steps towards a system that can help to estimate ice content in practical situations. We further advanced the acquisition system; one of the aims being a larger penetration depth. The 'Chameleon II' is a system for the broad-band spectral detection of complex electrical signal and for the usage under challenging field conditions. We also developed a method to estimate ice content from the electrical parameters that can actually be determined by the 2-D inversion approach.

To demonstrate the feasibility of the method, we carried out a field survey at the Shestakovka River Basin, near the Russian city Yakutsk. Investigations by Lebedeva et al. (2019) confirmed the presence of permafrost within the first few meters of depth and the occurrence of an unfrozen water-bearing talik within the frozen ground. The results will be compared with existing knowledge of the subsurface from a borehole and geophysical investigations.

## 2   Electrical permittivity of frozen soil

The complex electrical impedance of the ground is measured in terms of its magnitude $|Z|$ and phase shift $\varphi(Z)$ over a wide frequency range at several discrete frequencies. It contains the full information about the two material dependent properties of the ground: the electrical resistivity $\rho$ and the relative dielectric permittivity $\varepsilon_r$, being responsible for the behaviour of dissipation and storage of electromagnetic energy.

Many natural materials and material compositions show a frequency dependent behavior in electrical properties, due to polarization effects. Several processes leading to a polarizability are known, which can be distinguished by their strength and their occurrence in frequency range (Loewer et al., 2017). A mathematical description of those effects can be achieved by a parametrization of the frequency dependent permittivity. As discussed by Mudler et al. (2019), a useful parameterization for 2-D inversion is the model suggested by Cole and Cole (1941) extended by a third term including the DC conduction for a comprehensive description of the electrical behaviour. This extended Cole-Cole model (in the following just named Cole-Cole model), which has been used previously for cryospheric investigations (e.g. Bittelli et al., 2004; Grimm and Stillman, 2015), takes the following form in terms of the effective complex relative permittivity:

$$\varepsilon_r^* = \varepsilon_r' - i\varepsilon_r'' = \varepsilon_{HF} + \frac{\varepsilon_{DC} - \varepsilon_{HF}}{1 + (i\omega\tau)^c} + \frac{1}{i\omega\varepsilon_0\rho_{DC}}, \tag{1}$$

where $\varepsilon_r'$ and $\varepsilon_r''$ are the real and imaginary part of the complex relative dielectric permittivity, with the permittivity of free

space $\varepsilon_0 = 8.854 \times 10^{-12} F\ m^{-1}$, the angular frequency $\omega$ and the imaginary unit $i = \sqrt{-1}$. The five Cole-Cole parameters are: the direct current (DC) resistivity $\rho_{DC}$, the low-frequency limit $\varepsilon_{DC}$ and the high-frequency limit $\varepsilon_{HF}$ for the permittivity, the relaxation time $\tau$ and the relaxation exponent $c$. The equation represents the behaviour for low frequencies dominated by the resistivity, the behaviour for high frequencies controlled by the permittivity, and the part of relaxation process occurring in

between (Grimm and Stillman, 2015). In the case of pure ice, the polarization is caused by protonic defects, i.e. the Bjerrum defect (e.g. Hobbs, 2010). The relaxation process can be approximated by the Debye model, which is a special case of a Cole-Cole model with fixed exponent $c = 1$, resulting in a reduction of free model parameter to four (e.g. Petrenko and Whitworth, 2002; Artemov and Volkov, 2014).

In general, there is a choice whether the data interpretation is based on imaginary conductivity, or on the real part of permit-

tivity, because the two are mathematically equivalent. Whereas for low-frequency ($< 100\,\mathrm{Hz}$) SIP measurements, imaginary conductivity is often preferred (Loewer et al., 2017), for high-frequency SIP covering the relaxation of ice, permittivity is generally considered (Bittelli et al., 2004).

The strong relaxation process of ice occurs around a relaxation time $\tau$ of $2 \times 10^{-5}\,\mathrm{s}$ at $0°C$, shifting to longer times for decreasing temperature (Sasaki et al., 2016). Therefore, the detection of the ice relaxation requires frequency measurements in

the range of kHz. Our system was designed to measure in a range up to above $100\,\mathrm{kHz}$. In areas under periglacial conditions, due to high values of permittivity and resistivity of ice (Hauck and Kneisel, 2008), both mechanisms, the conduction and displacement currents, influence the signal of the impedance within the measured frequency range.

## 3   Ice content estimation

The HFIP method uses the same field procedures as DC resistivity methods. In principle, soundings and tomographic mea-

surements in different configurations, such as Wenner and Schlumberger, are available, but the dipole-dipole configuration is generally preferred because it is less sensitive to EM coupling effects.

The measured impedance spectra can be analyzed either separately, or in a two-dimensional inversion algorithm. Both procedures are described by Mudler et al. (2019) and use the Cole-Cole model (eq. 1) to fit the spectral data. The first approach can be used to evaluate single spectra to give an overview of the subsurface information or in case of homogeneous grounds

or samples, as in laboratory studies. To obtain the spatial distribution of electrical parameters, a 2-D inversion algorithm embedded in the geophysical modelling and inversion toolbox AarhusInv (Auken et al., 2014) was developed and presented by Mudler et al. (2019). The inversion leads to the distribution of the five Cole-Cole parameters defined above in eq. 1, which can be visualized in separate two-dimensional sections. Therefore, an interpretation of subsurface material and structure based on the full spectral information is possible.

Here, we add another step to the data analysis - the quantitative estimation of ice content. Several models for ice content estimation based on the frequency dependent electrical information of subsurface material exist in the literature, based on empirical approaches (e.g. Bittelli et al., 2004; Stillman and Grimm, 2010) and on physical models, such as the Maxwell-Wagner polarization, (e.g. Kozhevnikov and Antonov, 2012; Zorin and Ageev, 2017). The models have been developed theoretically

and have been tested with laboratory data of frozen samples. Under laboratory conditions, additional measurements, such as the saturation of the sample or the permittivity of the unsaturated matrix material can be performed. However, none of the existing theories or empirical relationships estimates the ice content using the five parameters which we can actually obtain from field measurements. Therefore, we suggest an additional inversion that is applied to the results of the spatial 2-D inversion and converts the five parameters into ice content. Our method is based on the theory of Zorin and Ageev (2017), which describes the subsurface material as a two-component mixture of ice and an ice-free part of the soil. In that theory it is assumed, that the polarization is fully caused by the ice fraction. The complex electrical conductivity represents a power mean of both components weighted by the ice content $\alpha$:

$$\tilde{\sigma}_b^k(\omega) = (1 - \alpha)\,\tilde{\sigma}_m^k(\omega) + \alpha\,\tilde{\sigma}_i^k(\omega) \tag{2}$$

where $\tilde{\sigma}$ are the complex conductivities ($\tilde{\sigma} = i\omega\varepsilon_0\varepsilon_r^*$) of the bulk soil (b), the ice (i) and the ice-free matrix material (m). As the ice-free part is assumed to be non-polarizable, its complex conductivity can be described by constant electrical parameters over frequency:

$$\tilde{\sigma}_m(\omega) = \sigma_m + i\omega\varepsilon_0\varepsilon_m \tag{3}$$

with $\sigma_m$ and $\varepsilon_m$ being the DC conductivity and the relative permittivity of the ice-free matrix. The matrix itself may be a composition of different materials, which are considered to be included in $\sigma_m$ and $\varepsilon_m$. Therefore eq. 3 describes an effective matrix conductivity. The possible presence of water is included in this fraction as well, because the electrical parameters of water are constant in the examined frequency range. Specifically, the polarization processes for water takes place at higher frequencies ($> 1\,\mathrm{GHz}$) (Artemov, 2019).

The frequency dependent behaviour of the ice signal can be described by a Debye model, formally equivalent to (eq. 1), written down for conductivity with $c = 1$:

$$\tilde{\sigma}_i(\omega) = \sigma_{i,DC} + i\omega\varepsilon_0\left(\varepsilon_{i,HF} + \frac{\varepsilon_{i,DC} - \varepsilon_{i,HF}}{1 + i\omega\tau_i}\right) \tag{4}$$

with $\sigma_{i,DC}$ being the DC conductivity of ice, $\varepsilon_{i,HF}$ and $\varepsilon_{i,DC}$ the low and high frequency limit of the relative permittivity of ice and $\tau_i$ the relaxation time for ice. The exponent $k$ in eq. 2 is assumed to reflect the spatial microstructure of both components, with $k \in [-1, 1]$. Therefore, eq. 2 can be considered a generalization of several mixing models, with specific values of $k$ corresponding to specific assumptions on the spatial distribution of two media. For a detailed discussion we refer the reader to Zorin and Ageev (2017), who describe which values of $k$ correspond to which assumption on the type of mixture and

previously published equations. Also note that eq. 2 implicitly includes Maxwell-Wagner polarization, which results from the contact between two media with different electrical properties. The generality of the model introduced by parameter $k$, and the simplicity introduced by describing the non-ice fraction by the matrix conductivity, are our main reasons to use this model for field data interpretation. The model has been validated by Zorin and Ageev (2017) on one laboratory data set, and it is expected

that it will be corroborated by further laboratory investigations.

Overall, eq. 2 comprises eight free parameters: the ice content $\alpha$, the formation exponent $k$, the two electric parameters of the ice-free part (eq. 3) and the four electric parameters of the ice fraction (eq. 4). Fortunately, the three electric parameters describing the relaxation of pure ice, are well known and can be fixed, namely the high and low frequency permittivity $\varepsilon_{i,HF}$ (3.2) and $\varepsilon_{i,DC}$ (93) and the relaxation time $\tau_i$ ($2.2 \times 10^{-5}\,\mathrm{s}$). This reduces the number of unknowns to five, which is equal to

the number of parameters we obtain from the field measurements. We can therefore proceed to obtain the five unknowns of the two-component model, ice content being one of them.

The minimization problem is being treated as a conventional non-linear least-squares inversion with upper and lower boundaries for some of the parameters. Specifically, we constrain $k$ in the areas of expected significant ice content in the interval between $-0.3$ to $0.5$ and use an upper limit for the ice content $\alpha$ of $50\,\%$. The inversion with boundary constraints is realized

with the MATLAB 2017b function lsqnonlin. Without using the parameter boundaries, the inversion tends to run into local minima in some cases and has difficulties to find the optimum model.

The procedure outlined above is based on the assumption that the polarization of ice is the dominant effect in the investigated frequency range, and that other mechanisms, such as polarization of the electric double layer (EDL), can be neglected

in the sense that they do not significantly distort ice content estimation if ignored. In order to justify this assumption and to illustrate the possible impact of EDL polarization, we tentatively expand our model (eq. 2) by the model suggested by Coperey et al. (2019). These authors simulate the EDL polarization of frozen porous media, explicitly ignoring the polarization of the ice itself, which is described by our eq. 2. They term the EDL polarization as "low frequency" defined by an upper limit at $10\,\mathrm{kHz}$. They also discuss conductivity spectra measured on different soil samples, where the polarization of the ice itself

dominates EDL polarization at frequencies above approx. $100\,\mathrm{Hz}$, depending on soil type and temperature.

We incorporate EDL polarization into the electrical conductivity of the effective matrix described by eq. 3 above. Specifically, we replace the real conductivity of the matrix $\sigma_m$ by the complex conductivity defined by eqs. 17 - 20 in Coperey et al. (2019). Furthermore, we follow their suggestion to apply the Drake model also known as "constant phase" model, in which the phase shift and quadrature conductivity are independent of frequency over a wide frequency range. Further details on the Drake

model can be found in Revil et al. (2017). The fluid water content of their equation is replaced by $1-\alpha$, where $\alpha$ is ice content. The extended eq. 3 then reads

$$\tilde{\sigma}_m(\omega) = \sigma_\infty + i\sigma'' + i\omega\varepsilon_0\varepsilon_m \tag{5}$$

with

$$\sigma_\infty = (1 - \alpha) \; [\phi \; \sigma_W(T) + \rho_g \; B(T) \; CEC] \tag{6}$$

and

$$\sigma'' = -(1 - \alpha) \; \frac{\rho_g \; \lambda(T)}{d} \; CEC. \tag{7}$$

In these equations, T is temperature, $\phi$ is porosity, $\rho_g$ is grain density, $\sigma_W$ is pore water conductivity and CEC is cation exchange capacity. Parameters B and $\lambda$ are apparent mobilities related to surface conduction and polarization associated with the quadrature conductivity (Revil et al., 2017). Parameter d is related to the number of frequency decades over which the Drake model is valid.

Figure 1 shows the result of a synthetic study, where complex conductivity is displayed in terms of its magnitude and phase shift vs. frequency. The parameters were chosen to correspond to a clayey soil according to table 1 in Coperey et al. (2019). The cation exchange capacity (CEC) which controls the magnitude of the EDL polarization is relatively large for clayey soils compared to other materials. The temperature-dependent parameters were chosen for a temperature close to the freezing point because this is likely representative for the conditions at the field site discussed further below.

The ice content is observed to control the phase shift and the conductivity magnitude over the entire frequency range, indicating that the HFIP measurements are well suited for a quantitative estimation. The effect of the EDL polarization is only visible for low ice contents and at low frequencies $< 100\,\mathrm{Hz}$. For these parameter ranges, the ice polarization is weak and the EDL polarization becomes increasingly important. In the frequency range which is essential for ice content estimation, between $1\,\mathrm{kHz}$ and $100\,\mathrm{kHz}$, the curves with and without EDL polarization do not visibly differ from each other.

Ignoring EDL polarization when estimating ice content seems justified for the parameter set chosen for the study. Since EDL polarization is further dependent on many parameters, a general statement may be difficult to derive. However, the most important parameter controlling EDL polarization is the CEC, which was chosen here to be relatively large to give a worst-case scenario. The second key parameter is frequency. Although in the Drake model, low frequency polarization itself is not frequency dependent, it becomes more relevant with decreasing frequency because ice polarization decreases. Moreover, EDL polarization may also be vary with frequency, depending on the dominant grain size of the media, with smaller grain size corresponding to higher frequencies (e.g. Leroy et al., 2008). Therefore, our assumption of negligible EDL polarization may fail under specific conditions, such as particularly large CEC and/or small grain sizes.

Under normal conditions, however, EDL polarization will only have a second order effect on ice content estimation. Therefore, in order to keep the number of unknown parameters as few as possible, we continue to use the simplified version (eq. 3). An extension to consider EDL polarization, or other polarization processes, or a quantitative investigation of the distortion might be subject of future work. In the present form, the model used here might be considered valid approximation in the frequency

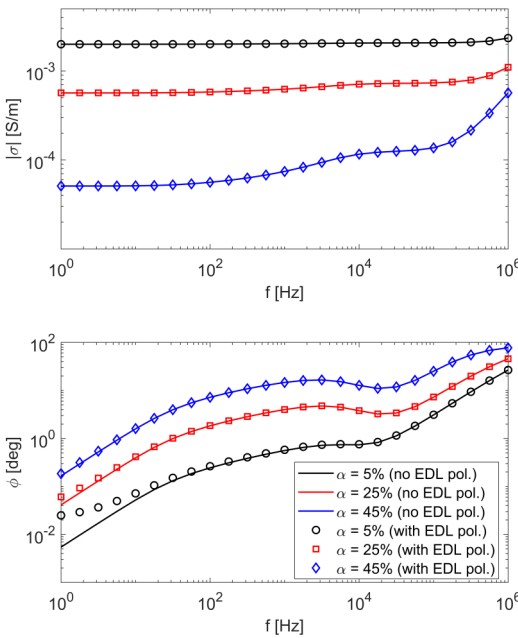

**Figure 1.** Magnitude (top panel) and phase shift (bottom panel) of the complex electrical conductivity vs. frequency for different ice contents $\alpha$. The lines denote the simulation without EDL polarization, calculated with eqs. 2 - 4, the symbols were calculated using the extended form of eq. 3 given by eq. 5. The parameters in eq. 5 are: $\phi = 0.5$, $\rho_g = 2650\,\frac{kg}{m^3}$, $\sigma_W = 0.1\,\frac{S}{m}$, $CEC = 18.8\,\frac{meq}{100g}$, $B = 3\cdot 10 - 9\,\frac{m^2}{Vs}$, $\lambda = 3\cdot 10^{-10}\,\frac{m^2}{Vs}$, $d = 8.5$, and the parameters of the ice relaxation are $\varepsilon_m = 20$, $\sigma_{i,DC} = 10^{-7}\,\frac{S}{m}$, $\varepsilon_{i,DC} = 92$, $\varepsilon_{i,HF} = 3.2$, $\tau_i = 2.2\cdot 10^{-5}s$, $c = 1$, $k = \frac{1}{3}$.

range between $100\,\text{Hz}$ and $100\,\text{kHz}$.

Another parameter that is not varied in our current model is temperature. Temperature dependence is one of the main aspects in the work by Coperey et al. (2019), and therefore becomes relevant as soon as EDL polarization itself becomes relevant. The high-frequency ice polarization also depends on temperature, which is well understood and described in literature, the main effect being a change of relaxation time in eq. 4 with temperature (Sasaki et al., 2016). The impact on ice content estimation might become significant if high accuracy is required and if temperature changes with time, e.g. during a monitoring experiment. For the field data discussed below, the temperature is close to the freezing point, and the measurements were taking within a few days, where temperature can safely be assumed to be stable. Furthermore, the change of relaxation time with temperature is relatively moderate. An extension of the model to include temperature dependence will be relatively straightforward, but at the same time increase the number of free parameters, a complication we avoid at this early stage of model development.

In the final step, the ice content model has to be applied to measured data. For a homogeneous sample, e.g. in the laboratory,

measured spectra can be directly fitted to eq. 2. In the case of heterogeneous field data, the distribution of electrical parameters within the subsurface has to be determined first. Since the inversion after Mudler et al. (2019) is currently the only existing method for two-dimensional interpretation of HFIP data, it is necessary to perform this process as the first step of data analysis. In our application, we use the inverted 2-D spectra as input for the fitting process for the ice content model after Zorin and
Ageev (2017) and thus obtain the two-dimensional distribution of ice content.

## 4   Chameleon II Instrument

Impedance measurements at the field scale with 4-point configurations up to frequencies sufficiently large to capture the ice relaxation pose special challenges on the hardware. As can be seen from figure 1, the peak frequency occurs around $10\,\mathrm{kHz}$, and it is desirable to obtain data even above that frequency to recover the full shape of the spectrum, which means that the
hardware should be able to measure up to $> 100\,\mathrm{kHz}$. Unlike quasi DC measurements, capacitive and inductive coupling effects occur at high frequencies, which, if not taken into account, lead to irreversibly disturbed measurement results. Most of the coupling occurs between the measuring cables themselves and between them and the ground. In principle, inductive coupling can be calculated with suitable simulation programs. In addition to the position of the current and potential electrodes, the known course of the measuring cables must be taken into account. The situation is different with capacitive coupling effects.
Here, quantitative modelling is complicated by the fact that the strength of the coupling is determined, among other factors, by the coupling resistances of the two current electrodes, as well as the distance of the current cables from the ground. Both are usually not known with sufficient accuracy. Capacitive coupling must therefore a priori be avoided as far as possible during the measurement with the help of a suitable measurement concept. Two-point measurements, which are sometimes used for high-frequency measurements in the laboratory to avoid coupling effects (e.g. Volkmann and Klitzsch, 2015), are generally not
feasible at the field scale, because they are only sensitive to the immediate vicinity of the electrodes (e.g. Hördt et al., 2013). We drew on experience with a prototype "Chameleon-I" (Radić, 2013) which we optimized in several ways. First, in order to achieve larger penetration depths, we increased transmitter power and took additional measures to reduce the stronger coupling effects associated with longer measurement lines. Figure 2 shows the block diagram of the Chameleon-II instrument which is designed for measurements up to $230\,\mathrm{kHz}$. An essential innovation is that a complete transmitter is now positioned at each
of the two current electrodes. In contrast to the previous instrument, this results in a symmetrical current dipole. Both transmitters are connected in series via a power cable. As a result, the two transmitter voltages add up to $\pm 800$ volts. Thanks to the symmetry of the current dipole, there is no more need for lossy shielding of the current cable as with the Chameleon-I. In practice, the current electrodes have unequal contact resistances. The result is a potential difference between the current cable connecting the two transmitters and the ground below. At the high measurement frequencies used, this causes capacitive
leakage currents, which lead to systematic measurement errors. The presence of leakage currents is detected by comparing the current strengths measured in the transmitters, directly at the current electrodes. The compensation is done by choosing individual output voltages for the two transmitters, such that the two current strengths equalize and a measurement error caused by leakage current is minimized. This procedure to compensate the coupling between the current cables (C) and the earth (E)

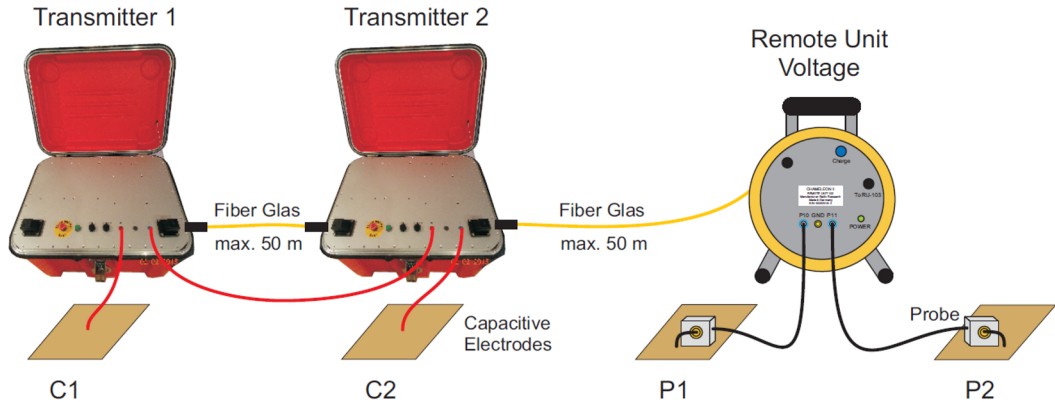

**Figure 2.** Arrangement of the components of the Chameleon II measuring system for capacitive and galvanic impedance measurements. The red lines are analog current cables. There is one transmitter at each current electrode. The remote unit, integrated in a cable drum, is connected to the potential electrodes via active probes and potential cables (black). Optic fibre cables (yellow) are used to avoid coupling effects. (Radic Research: Flyer Chameleon II)

during the measurements is called "CEC" (Radić and Klitzsch, 2012) within this context. Undesirable potential differences also occur to a lesser extent between the potential cables and the ground below. To minimize the negative impact on the voltage measurement, active probes are used directly at the potential electrodes. During fieldwork, special care is taken to keep EM coupling effects under control. In particular, any analogue cable is kept as short as possible, e.g. by placing the probes directly

at the electrodes, and by using current cables of the exact length required.

In addition to the EM distortion resulting from the direct coupling of the cables with each other and to the ground, there is also the effect of induced currents in the ground, which can not be avoided. Mudler et al. (2019) suggested equations based on the theory developed by Weidelt (1997) to estimate whether induction effects can be ignored. For the field site discussed below, assuming subsurface resistivities higher than $2000\,\Omega\mathrm{m}$ for the top layer, according to fig. 5 and Lebedeva et al. (2019),

our estimation reveals that induction effects are small enough to be neglected over almost the entire measuring range. For the highest frequencies and largest configurations, the conditions for neglecting induction are no longer strictly fulfilled and might have to be considered explicitly by modelling in future work.

Quantitatively, we may consider the data errors as a sum of random errors and systematic errors. The random errors can be assessed by repetitive measurements or by the statistics used to determine the impedance magnitude and the phase shift from the

time series. These are provided by the instrument, and depend on many factors, but are typically below $10\,\%$ for the impedance magnitude and below 5 degrees for the phase shift (at the high frequencies around $10\,\mathrm{kHz}$). For decreasing measurement frequencies, these errors also tend to decrease. The main source of systematic errors is likely the EM coupling effects discussed above, which are more difficult to evaluate quantitatively. Here, we use two methods to approach a quantitative assessment. First, we compare a data set with and without the CEC procedure from the HFIP profile of the present study (fig. 3a). The

average difference in the impedance magnitude is below $1\,\%$ for the magnitude, and $< 1$ degrees for the phase shift, indicating

that the coupling from the transmitter cable is not significant in this particular case.

The second method consists of reciprocal measurements, which are generally considered a good measure of data error (Ramirez et al., 1999; Flores Orozco et al., 2012). In reciprocal measurements, the roles of transmitter and receiver are interchanged, which should theoretically provide identical results if no systematic errors are present. The difference between original and reciprocal data can therefore be taken as an error estimate. Unfortunately, we have not carried out reciprocal measurements during the survey discussed here. Therefore, we resort to reciprocal measurements carried out during a later survey to provide a general assessment of the Chameleon-II performance, rather than a specific assessment of data errors for the Yakutsk survey. The data are from an alpine permafrost site at area Cervinia, Italy, which was described by (Mollaret et al., 2019). Our measurements with Chameleon-II were taken in summer 2021. An example of reciprocal spectra from that survey is shown in fig. 3b. Again, the errors are below $5\%$ and 1 degrees for magnitude and phase shift, respectively. The few percent difference in the magnitude may be explained by the fact that a dipole-dipole configuration was used, which has a relatively low signal strength compared to other configurations with the same depth of investigation. Moreover, the conditions for electrode coupling were relatively harsh in the sense that the surface was rocky and irregular. Although the errors depend on the conditions at a specific site and are not necessarily transferable to the data discussed here, we conclude that there seem to be no major instrumental issues that might spoil ice content estimation.

Both instruments (Chameleon-I and Chameleon-II) can be used with capacitive electrodes, namely the Capacitively Coupled Resistivity method (CCR), applied by Przyklenk et al. (2016) and Mudler et al. (2019). Impedance measurements using conventional galvanic electrode coupling can be applied as well. As the logistical advantages of capacitive coupling were not significant at our field site in Yakutia, measurements were conducted using galvanic coupling.

## 5 Field site and additional investigations

The Shestakovka River Basin in Central Yakutia, Russia, is a field site of the Melnikov Permafrost Insitute (Yakutsk). The area is located in the continuous permafrost zone. The field site and its geological features have been described by Lebedeva et al. (2019). The area is partially covered by pine forest and larch forest. Swamp-like zones occur along the creeks. Sandy deposits dominate the top layers of the geological cross section. The thickness of permafrost in the area can be up to several hundred meters with an overlaying shallow active layer varying from $0.5\,\mathrm{m}$ to a maximum of $4\,\mathrm{m}$. The research of Lebedeva et al. (2019) focused on the occurrence of suprapermafrost subaerial talik and subsurface thermal anomalies that results in local unfrozen, water-bearing areas within the permafrost. Their results from borehole analysis and geophysical measurements clearly indicate talik within the first few meters depth. The phenomenon is known to exist in Central Yakutia and in particular in the area of Shestakovka River Basin.

The location of the study area is presented in fig. 4. The ERT profile is an extension of the measurements from Lebedeva et al. (2019), named 'profile III'. In that context, the results of well 3/16-P were already discussed. That ERT profile was extended by $250\,\mathrm{m}$ in the north-east direction and now has a length of approximately $500\,\mathrm{m}$. On this profile extension, a new borehole

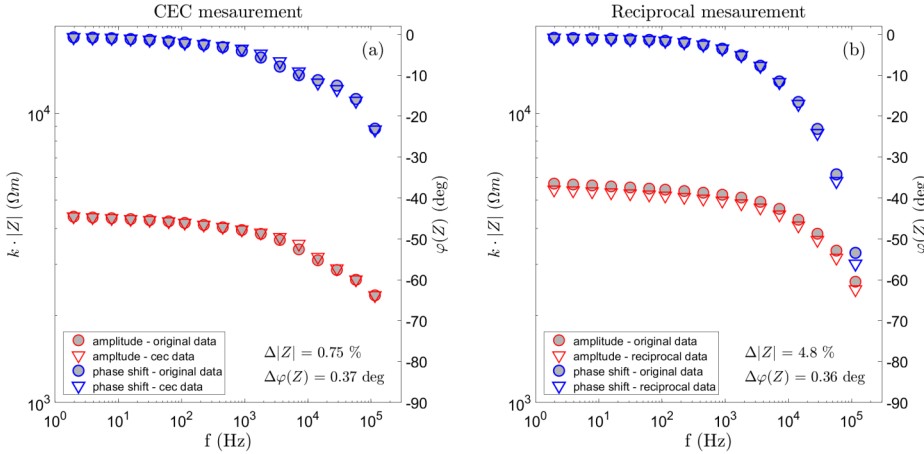

**Figure 3.** Panel a: Impedance magnitude and phase spectra from the survey at Yakutia, recorded with and without CEC compensation. The data were measured in a dipole-dipole configuration with $a = 2\,\mathrm{m}$ and $n = 8$. Panel b: Impedance magnitude and phase spectra from a survey at an alpine permafrost site. The data were measured with $a = 2\,\mathrm{m}$ and $n = 5$ in original and reciprocal dipole-dipole configuration. The amplitude and phase data are shown in red and blue colors, respectively.

was drilled, named 'well 3/18'. The two-dimensional resistivity distribution from the ERT profile is displayed in fig. 5 for the depth range up to $10\,\mathrm{m}$. While the right half of the profile was already part of the investigations in Lebedeva et al. (2019), the result of the left part of the profile is presented for the first time. It can be seen that the previously known areas of lower resistivity ($< 3000\,\Omega\mathrm{m}$) also extend on the profile extension (left half), albeit somewhat less pronounced. In these near-surface

5 areas, thawed deposits within the permafrost, i.e. the talik areas, occur. Moreover, a highly resistive zone ($> 8000\,\Omega\mathrm{m}$) between profile meters 100 and 300 could indicate a higher ice content within the permafrost compared to the surrounding areas. The ERT results show that the talik is not a continuous layer, but is separated into water bearing channels orientated along the slope towards the river (Lebedeva et al., 2019).

Focusing on the subsurface structure of the first $10\,\mathrm{m}$ of depth, being the investigation area of the applied HFIP method, fig. 6

10 shows the results of the lithology, the soil moisture and the resistivity of the investigation from well 3/18. The subsoil consists of deposits of fine sand starting below a thin top-soil layer. The thawed layer occurs between $2.2\,\mathrm{m}$ and $7.7\,\mathrm{m}$ depth, bounded above and below by permafrost. This structure is very similar to the borehole investigations from well 3/16-P, presented in Lebedeva et al. (2019). In order to obtain more accurate estimates of the thickness of the near-surface unfrozen layer, separate measurements using a dipstick which penetrates the ground to the frozen layer during the time of the HFIP measurements were

15 undertaken, showing a thickness slightly less than $50\,\mathrm{cm}$ along the HFIP profile. The soil moisture exhibits values up to around $20\,\%$ above the talik, increasing towards its upper boundary. The talik area itself is fully saturated. Since only frozen cores were sampled from the borehole, data of the soil moisture are available only for the corresponding depth range.

In May 2019, we performed a field survey with the HFIP method using the Chameleon-II device. Here, we focus on a line profile of $50\,\mathrm{m}$ length, the position of which is also shown in fig. 4 (B to B'). The HFIP profile is defined south-west directed

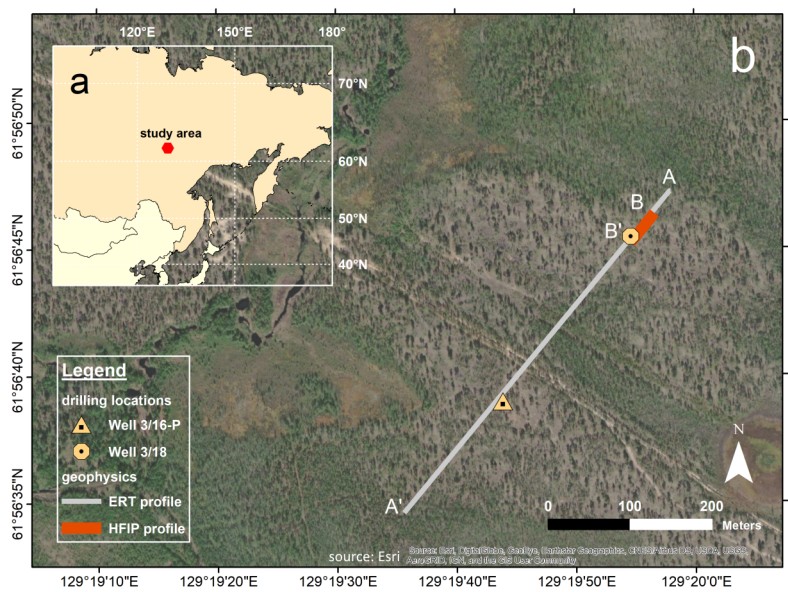

**Figure 4.** Geographical maps of the field site. Panel a gives an overview of the location of the study area in Russia. In Panel b the position of the ERT profile, directed from A to A′, and the HFIP profile from B to B′ are shown, as well as the locations of boreholes 3/16-P and 3/18.

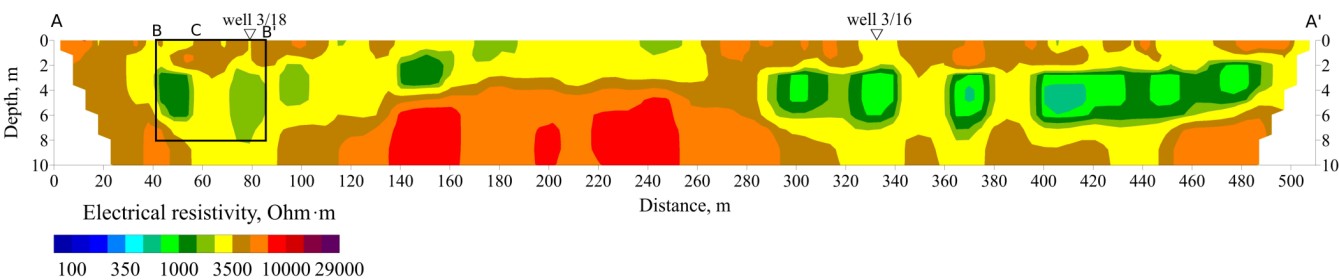

**Figure 5.** Two-dimensional section of the electrical resistivity over the whole ERT profile of around $500\,\mathrm{m}$ length (A to A′). The black rectangle marks the corresponding subsurface area of the HFIP measurements (B to B′), including the midpoint location of the HFIP sounding (C) and the location of well 3/18.

and starts around $40\,\mathrm{m}$ north-east of well 3/18. The measurements were carried out with a dipole-dipole configuration, with a dipole length of $1.5\,\mathrm{m}$ and where the dipole spacing varied between $1.5\,\mathrm{m}$ and $42\,\mathrm{m}$. Overall, more than $100$ impedance spectra were measured, with a frequency range from $2\,\mathrm{Hz}$ to $115\,\mathrm{kHz}$. Additionally a separate sounding in Schlumberger configuration with a maximum spacing of $\overline{AB} = 32\,\mathrm{m}$ was performed on the profile, at the location marked with C in fig. 5.

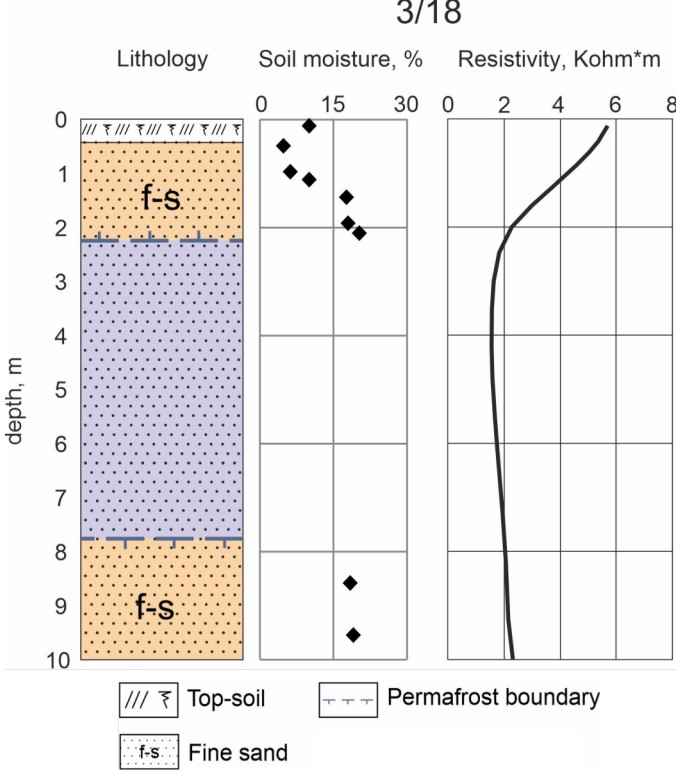

**Figure 6.** Results of measurements in well 3/18 - the lithology, soil moisture and resistivity are displayed from left to right. The drilling and investigations were conducted in April 2018. The blue area in the lithology section marks the unfrozen talik between the frozen areas displayed in yellow.

## 6    Results: the spectral signal of permafrost

A Schlumberger sounding was carried out with the midpoint location given in fig. 4. Since enlarging electrode distance corresponds with higher investigation depth (e.g. Militzer and Weber, 1985), the spectra should give a first overview of the depth distribution of permafrost. Since the method is still in its infancy, and only few case histories exist, we find it useful to discuss raw data and investigate the direct impact of the presence of ice in the measured spectra. The purpose is to illustrate how field data in a quasi-layered situation in a permafrost environment can look like and how the data varies with investigation depth.

Two data sets, one for the smallest spacing ($\overline{AB} = 2\,\mathrm{m}$) and one for an intermediate spacing at $\overline{AB} = 12\,\mathrm{m}$, are shown in fig. 7. Each symbol corresponds to the magnitude and the phase shift of the impedance $Z$ respectively for a discretely measured frequency. The phase shift is shown up to $115\,\mathrm{kHz}$, however, for the magnitude, values are only displayed until $60\,\mathrm{kHz}$, as the highest frequency values contain high errors. Note that the phase shift is just shown down to $-20°$, and not the full range for the phase shift of $-90°$.

The spectral signal of $\overline{AB} = 12\,\mathrm{m}$ shows a strong frequency dependence between $10^2 - 10^5\,\mathrm{Hz}$ with a clear peak, representing

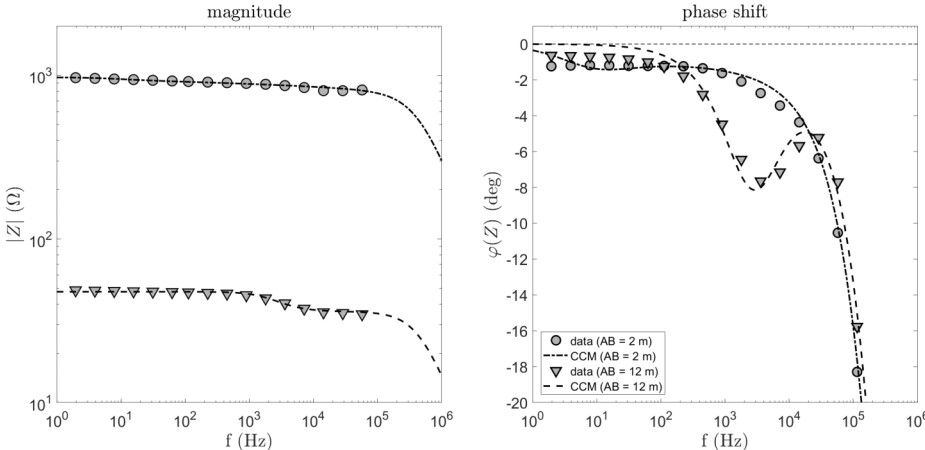

**Figure 7.** Spectra of the magnitude (left) and the phase shift (right) of the impedance $Z$ for two measurements with electrode spacing $\overline{AB} = 2\,\text{m}$ (dots) and $\overline{AB} = 12\,\text{m}$ (triangles). The larger spacing, associated with higher investigation depth, shows a distinct frequency peak above $1\,\text{kHz}$, whereas the other spacing shows weaker frequency dependence. The symbols indicate the measured data, the dashed lines are the data calculated for the single Cole-Cole model, the parameters of which are provided in table 1.

a single relaxation process. The data can be fitted by the Cole-Cole model (eq. 1), which provides the apparent parameters of the subsoil for every measured impedance spectra. The model is fitted to both amplitude and phase spectrum at the same time. The missing high-frequency magnitudes do not affect the resolution of the parameters as the phase peak, which is required to resolve the relaxation process, is well captured by the good quality data. The spectrum of the smallest spacing ($\overline{AB} = 2\,\text{m}$) is distinctly different and does not show any clearly visible frequency dependence corresponding to the frequency behaviour of the ice relaxation process. The decreasing magnitude and phase shift for high frequencies is due to the overlap between conduction and displacement current mechanism. Fitting the data of the smallest spacing by the Cole-Cole model, the result for the relaxation time is around $10^{-2}\,\text{s}$, much longer than expected for ice. Our interpretation is that the effect of ice is negligible or even not present for this electrode spacing, because of the unfrozen active layer at shallow depth. Accordingly, the spectra of this spacing can also be fitted by excluding the relaxation term in eq. 1, and by using only two parameters, which are both independent of frequency: a DC resistivity value and a high-frequency permittivity. The resulting model parameters are quite similar to the corresponding ones of the Cole-Cole model (tbl. 1). The apparent Cole-Cole parameters obtained by the fitting process of all spacings are listed in table 1. Except for the smallest spacing, all model spectra show a relaxation process around $5 \times 10^{-5}\,\text{s}$ which supports the assumption of ice relaxation. A closer look reveals systematic variations with increasing investigation depth. From the second smallest spacing on, both permittivity values are first increasing with depth and then decrease again towards the largest spacing. The relaxation time varies within a narrow range and decreases continuously with depth. The resistivity values increase for the first spacings and then decrease to the largest one. The model exponent $c$ does not show consistent variation but its proximity to one supports the hypothesis of ice causing the relaxation, since it is known that ice relaxation can be described by the Debye model ($c = 1$) (e.g Petrenko and Whitworth, 2002). The percentage rms values for

**Table 1.** Model parameters of the data fit for the different spacings of the sounding. The measured impedance spectra were fitted by the single Cole-Cole model (eq. 1) by five parameters. Additionally, for the smallest spacing another data fit with constant electrical parameters over frequency was made. The data fit is quantified by the root mean squares.

| spacing $\overline{AB}(m)$ | $\rho_{DC}(\Omega m)$ | $\varepsilon_{DC}$ | $\varepsilon_{HF}$ | $\tau(s)$ | $c$ | $rms_{|Z|}(\%)$ | $rms_{\varphi}(mrad)$ |
|---|---|---|---|---|---|---|---|
| 2 | $2.29 \cdot 10^3$ | 52441 | 21.6 | $2.4 \cdot 10^{-2}$ | 0.84 | 0.29 | 6 |
|   | $2.13 \cdot 10^3$ | – | 21.2 | – | – | 0.57 | 13 |
| 4 | $2.95 \cdot 10^3$ | 428 | 17.8 | $7.0 \cdot 10^{-5}$ | 0.96 | 0.36 | 13 |
| 6 | $3.15 \cdot 10^3$ | 492 | 17.5 | $5.3 \cdot 10^{-5}$ | 0.98 | 0.41 | 13 |
| 8 | $3.02 \cdot 10^3$ | 537 | 18.0 | $5.2 \cdot 10^{-5}$ | 0.99 | 0.41 | 12 |
| 12 | $2.61 \cdot 10^3$ | 705 | 19.4 | $5.1 \cdot 10^{-5}$ | 0.99 | 0.47 | 11 |
| 16 | $2.28 \cdot 10^3$ | 699 | 21.1 | $5.1 \cdot 10^{-5}$ | 0.99 | 0.52 | 9 |
| 20 | $2.18 \cdot 10^3$ | 607 | 21.4 | $4.7 \cdot 10^{-5}$ | 0.99 | 0.62 | 9 |
| 26 | $2.18 \cdot 10^3$ | 525 | 19.1 | $4.2 \cdot 10^{-5}$ | 0.99 | 0.92 | 14 |
| 32 | $2.14 \cdot 10^3$ | 512 | 16.2 | $4.1 \cdot 10^{-5}$ | 1.00 | 0.97 | 12 |

the magnitude spectra and absolute rms in radiant for the phase spectra are small for all spacings, indicating that the Cole-Cole model is well suited to describe the measured spectra.

Since the method is not yet well established, and the behaviour of the data in the presence and absence of ice is not generally known, we discuss the phase shift spectra in fig. 8. The left part corresponds to the spacings from $\overline{AB}$ 4 to 12 (fig 8a), the right one from $\overline{AB}$ 12 to 26 (fig 8b). All spectra show a characteristic peak with its minimum around a frequency of $10^4$ Hz. However, for increasing investigation depth, the intensity of the relaxation, characterized by the minimum phase shift, rises up gradually to $\overline{AB} = 12$ and further on decreases with higher investigation depth to $\overline{AB} = 26$. From the equation by Zorin and Ageev (2017) for a mixture of materials including ice it can be expected that as the ice content increases, so does the strength of the polarization, measured by the intensity of the peak in phase shifts. Modelling of pure ice, using the Debye relaxation (eq. 4) can result in a strong phase peak ($< -70°$) depending on the model parameters, in particular on the resistivity. Such strong phase peaks were indeed observed during field measurements (unpublished) with the Chameleon II device on alpine glacier ice, where values for the phase peak down to $-80°$ were measured. For a typical two-layer case of permafrost, with an unfrozen active layer above the frozen layer, measurements should show an increasing phase peak with investigation depth. This expected behaviour agrees well with calculated data in fig. 8a. The reduction of the phase peak for even greater spacing indicates a decrease of ice content with depth (fig. 8b). Our a priori knowledge about the existence of talik in the permafrost layer at the field site from the ERT and borehole investigations (fig. 5 & 6), points at an unfrozen layer as the most likely explanation.

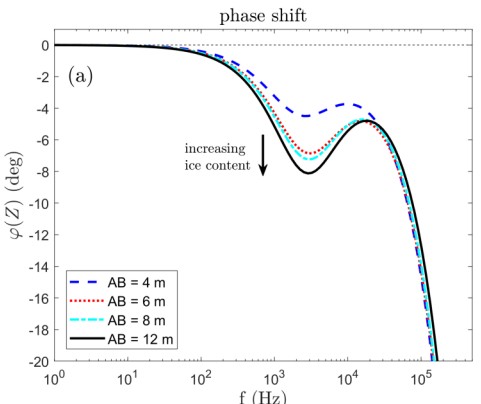
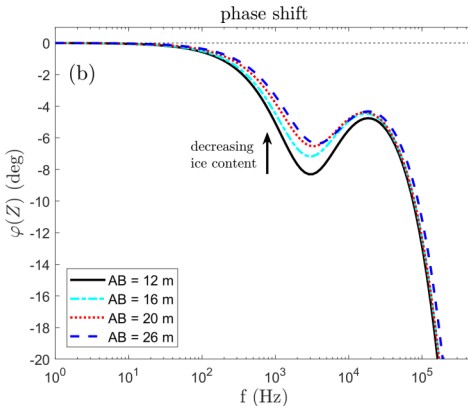

**Figure 8.** Calculated phase shift spectra for the Cole-Cole models used to fit the Schlumberger sounding data for the spacings $\overline{AB} = 4 - 12\,\text{m}$ (left) and for $\overline{AB} = 12 - 26\,\text{m}$ (right). The model fit of the data $\overline{AB} = 12\,\text{m}$ (black curve) is shown in both panels. All curves show a strong frequency dependence between $10^2 - 10^5$ Hz, whereby the intensity of phase peak gradually increases up to the spacing $\overline{AB} = 12\,\text{m}$ and decreases again for larger spacings, indicating an increase and decrease of ice content.

## 7    Results: Ice Content Estimation

A spectral 2-D inversion was carried out using the inversion tool AarhusInv (Auken et al., 2014). The procedure provides a spatial distribution of the five Cole-Cole parameters defined by eq. 1, and the result is shown in fig. 9. Brighter areas in the lower parts of each section are those regions where the maximum depth for reliable parameter estimation based on the depth-of-investigation estimate by (Fiandaca et al., 2015) is exceeded. The full spectral inversion algorithm calculates the distribution of the model parameters simultaneously over all frequencies and electrode configurations. The misfit of the inversion is calculated by the weighted mean square error over all inverted spectra, denoted by the symbol $\chi$ (Fiandaca et al., 2013). For the result shown in fig. 9, the misfit is $\chi = 1.9$.

The resistivity (panel a) is the same parameter as determined by other electric or electromagnetic methods in geophysics, for example by ERT, and has the same information content. The image is characterized by a highly resistive layer above a depth of $2\,\text{m}$. Underneath, two areas of low resistivity occur, where the one on the left half of the profile has a larger extension than the one on the right half. The areas are interpreted as water-bearing, unfrozen talik, according to (Lebedeva et al., 2019). The two talik zones are embedded in a background resistivity of around $3000\,\Omega\text{m}$ and seem to be separated around profile coordinate $25\,\text{m}$. Directly under the surface, a very shallow layer of lower resistivities is present, which may indicate a thin unfrozen active layer. Separately performed direct measurements of the unfrozen surface area during the days of geophysical measurements revealed an unfrozen layer thickness in the range of $30 - 50\,\text{cm}$ along the profile.

The other model parameters in fig. 9 show significant spatial variation as well, although the supposed talik areas are not as clearly visible as for the resistivity. The values for the model exponent $c$ (panel e) are close to 1, and only deviate for the supposedly unfrozen areas of talik and within the first meter. This is consistent with the fact that the Debye relaxation model

($c = 1$) describes the frequency dependence of ice, and $c = 1$ could therefore be an indicator for the location of frozen ground. The low frequency permittivity (panel b) and the relaxation time (panel d) exhibit a layered structure, with a distinct layer in about $1 - 3\,\mathrm{m}$ depth. The relaxation time for this layer is in the range of literature value for ice relaxation close to the melting point, which is between $2 \times 10^{-5} s$ and $2.2 \times 10^{-5} s$ (e.g. Auty and Cole, 1952; Artemov and Volkov, 2014; Sasaki et al., 2016).

The high frequency permittivity (panel c) has higher values within the first meter in depth and at an anomaly at the end of the profile. Below $1\,\mathrm{m}$, $\varepsilon_{HF}$ jumps to significantly smaller values. As the high-frequency permittivity of ice is characteristically low ($3 - 4$), areas of low values may indicate significant ice content.

The result for the DC resistivity parameter (panel a) is in good agreement with the corresponding ERT 2-D result, which is displayed in fig. 10. The different ways to obtain resistivity, i.e. different electrode geometries, inversion algorithms and most

importantly the use as a parameter in frequency-dependent conductivity function (in case of HFIP) vs. the direct measurement (ERT), causes differences in the images, but the overall structure is the same. The two separated low resistive areas are present in both images, as well as the highly resistive layer above. The boundary between those layers is consistent with the results obtained in well-3/18 (fig. 6), indicating the transition from permafrost to the unfrozen area in $2.2\,\mathrm{m}$ depth at the borehole location. The shallow surface layer is only visible in the resistivity result of the HFIP due to the higher resolution compared to

the large ERT profile.

It has to be mentioned that the 2-D inversion algorithm fits a Cole-Cole relaxation model for all areas of the subsurface. The relaxation process is dominated by that of ice, and if the data show no (ice) relaxation process, some of the five parameters become poorly constrained, which may lead to unrealistic values. Therefore, fig. 9 probably should not be interpreted in terms of anything else but ice relaxation and ice content. Tuning the inversion to allow for other relaxation processes to become

visible will be an issue for future developments of the inversion algorithm.

Within a next step, we transfer the frequency dependent information of the Cole-Cole model distribution into a value of ice content. For this purpose, the two-component model (eq. 2) after Zorin and Ageev (2017) was applied to the results of the 2-D inversion. For every cell of the inversion grid, the ice content is determined using the procedure described in chapter 3. The result is presented by the 2-D sections of the ice content $\alpha$ and the structural parameter $k$ in fig. 11. Overall, the distribution

of ice content gives a coherent picture, consistent with the expectations: Near the surface, a distinct layer with non-zero ice content between approx. $1\,\mathrm{m}$ and $2.5\,\mathrm{m}$ depth is embedded between layers of zero or low ice-content below $10\,\%$. At greater depths below $6\,\mathrm{m}$, the ice content increases again. The near-surface layer with zero ice-content corresponds to the currently unfrozen part of the active layer. The distinct layer with highest ice content is the frozen part of the soil with significant moisture, whereas the zones with zero ice content between $3\,\mathrm{m}$ and $7\,\mathrm{m}$ depth correspond to the unfrozen talik.

The borehole information on frozen and unfrozen state, in combination with the soil moisture measurements (fig. 6) even allows a quantitative assessment of the estimated ice content. The soil moisture in the frozen sections of the borehole can be used as an upper limit of ice content. It is not identical to ice content, however, since even at temperatures below $0\,^{\circ}\mathrm{C}$, a substantial amounts of water is still unfrozen. The amount of unfrozen water depends on lithology, salinity and temperature (Watanabe and Mizoguchi, 2002). Since these parameters are not known with sufficient accuracy, we consider the soil moisture

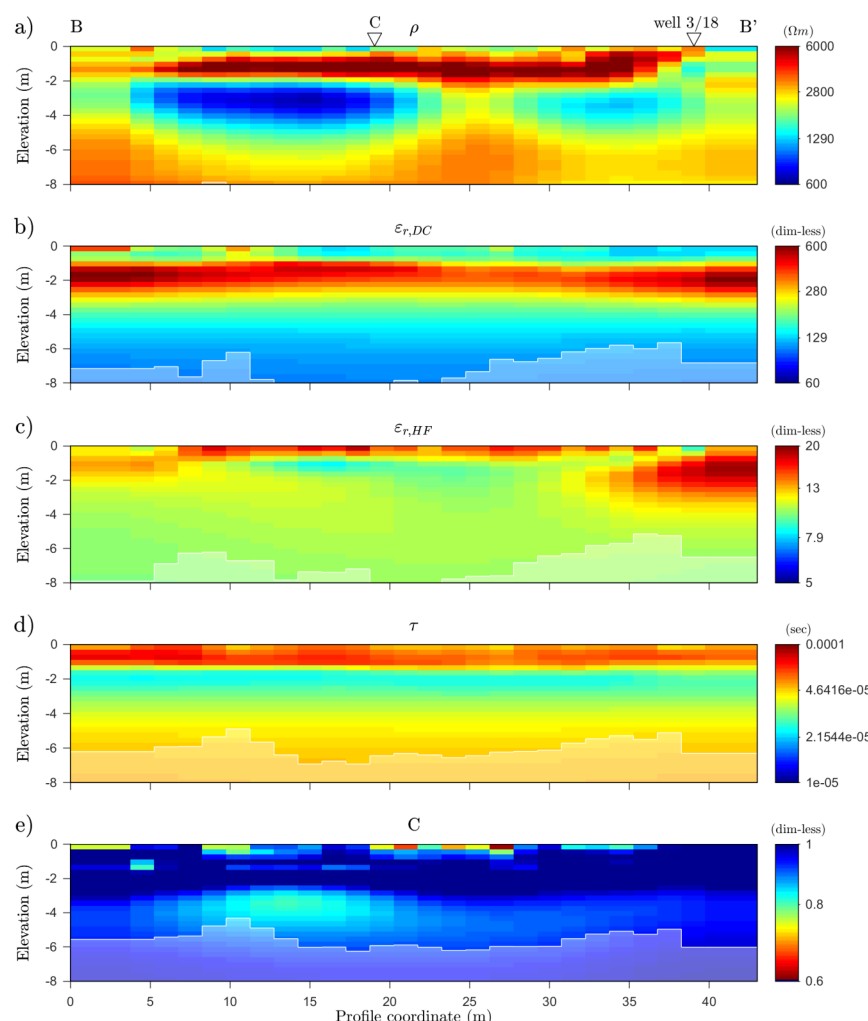

**Figure 9.** Two-dimensional inversion results for the profile of HFIP measurements at the Shestakovka River basin (B to B′), separately shown for each of the five Cole-Cole parameters (a-e). The total data fit is $\chi = 1.9$. Brighter areas mark zones of decreased reliability based on the depth of investigation (Fiandaca et al., 2015). The locations of the Schlumberger sounding (C) and the well 3/18 are marked additionally at panel a.

as an upper boundary for the ice content to compare our results. Just above the talik, in approx. $2.2\,\mathrm{m}$ m depth, the soil moisture reaches values of more than $20\,\%$ with an increasing tendency with depth. Close to the boundary to the talik, the ice content may be expected to be almost as high as in the unfrozen talik itself. The water content within the talik was not measured for well 3/18, but investigations by Lebedeva et al. (2019) indicate contents of $30\,\%$ and more. Therefore, it can be expected that the ice content just above the talik can reach values larger than $20\,\%$, or even up to $30\,\%$. The values of the HFIP-estimated ice content at the location of well 3/18 (fig. 11) are in the range of $30\,\%$, which means that HFIP seems to slightly overestimate

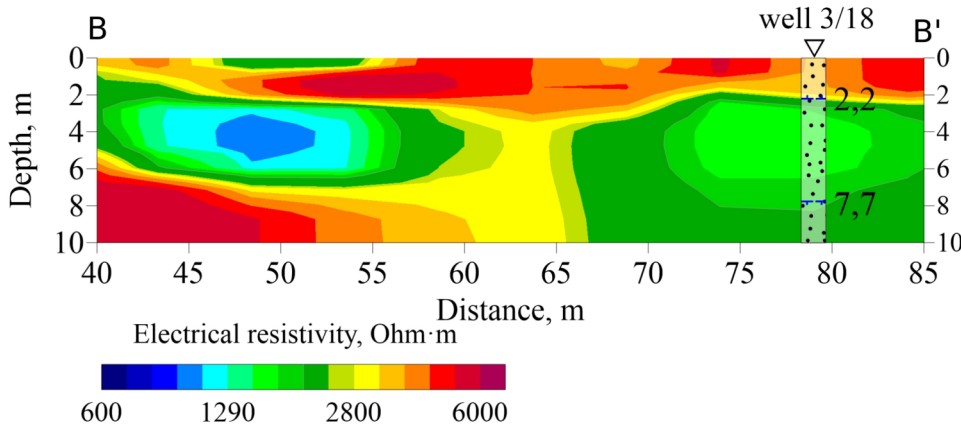

**Figure 10.** Two-dimensional electrical resistivity as results of the ERT inversion. Displayed is the part from profilemeter 40 to 85 of the bigger section in fig. 5 that corresponds to the section of the HFIP inversion in fig. 9 (B to B′). Additionally, the location of well 3/18 and its estimated boundaries from permafrost to the unfrozen layer are shown.

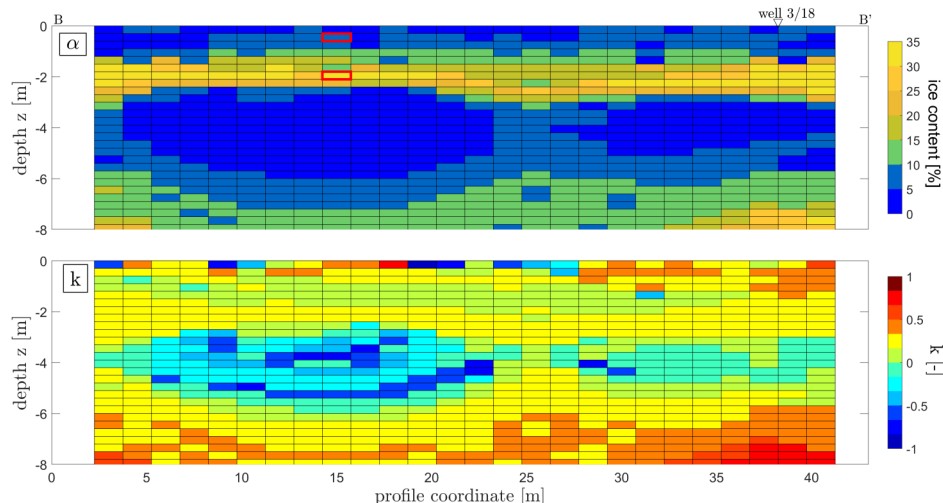

**Figure 11.** Two-dimensional sections of the percentage ice content $\alpha$ and the parameter $k$, calculated from the model after Zorin and Ageev (2017) using the procedure described in chapter 3. The displayed section is the same as in fig. 9 (B to B′). The two cells outlined in red in the upper section indicate the corresponding location of the spectra in fig. 12.

ice content when considering that soil moisture constitutes an upper limit. Over the entire profile, ice content values up to $35\%$ are reached which can be considered realistic.

The lower boundary of the talik at the location of well 3/18 is approx. at $7.7\,\mathrm{m}$ depth (fig. 6). In the HFIP results (fig. 11), the ice content at the location of well 3/18 increases to values above $10\%$ in slightly shallower depths, around $6\,\mathrm{m}$. However,

temperature measurements in well 3/18 indicate that the boundary to the frozen zone ($< 0.1\,°C$) actually moves and can vary between $5\,m$ and $9\,m$. If we also consider that the inversion algorithm uses a regularization and there is a natural limit to the accuracy at which the depth of boundaries can be determined, the HFIP result can be considered consistent with the borehole information. Due to the decreasing spatial coverage at greater depths, it can be assumed that the reliability of parameter resolution also decreases in the range below the depth of around $7\,m$.

The second parameter shown in fig. 11 is the exponent $k$. The parameter varies within a range from $0$ to $0.5$ over large areas of the 2-D section. This is especially the case for the ice bearing areas above and below the talik and is thus consistent with the considerations of Zorin and Ageev (2017), who identify this range to be expected for most practical cases for isotropic or moderately anisotropic media. The only areas where the values clearly deviate from this range and show negative values down to almost $-1$ are the identified talik areas and occasionally cells within the near surface active layer. A value of $-1$ is theoretically possible and would occur for the conductivity perpendicular to a strongly anisotropic layering (Zorin and Ageev, 2017). However, since these areas are ice-free, it is more likely that the two-component model is underdetermined. As a result, the parameter $k$ is poorly resolved and not meaningful.

For a more detailed analysis of the fitting process, fig. 12 exemplarily shows the spectra for two cells of the spatial grid, marked by red rectangles in the ice content section (see fig. 11). The model curve of the two-component model by Zorin and Ageev (2017) is shown for a cell with low estimated ice content (blue line), which lies in the range of the active layer, and one cell in the area of the frozen layer with maximum ice content (green line). The dashed lines are the corresponding Cole-Cole model curves to which the two-component model has been fitted. In the case of the active layer cell, the different models curves are compatible with each other, whereas discrepancies between the models in the high ice content region are evident, including the frequency range of the characteristic phase peak. Apparently, the frequency of the spectral peak can be well matched and the size of the peak can be approximated, but the exact curve shape seems to be different between the Cole-Cole model and the two-component model.

To obtain an overview over the entire section, we display the misfit in terms of root mean square (rms) values and displayed for magnitude and phase shift spectra in fig. 13, in the same fashion as the ice content itself. Except for a few outliers, the rms values are below $15\,\%$ for the magnitude and below $0.1\,rad$ for the phase shift. For depth ranges lower than $1\,m$ and greater than $2.5\,m$, the data fitting errors are small. In contrast, for the area in between, which represents the layer of significantly higher ice content above the talik, both rms panels show higher values. The increasing fitting errors for large ice content indicate that eq. 2 might no longer match perfectly with the Cole-Cole model. One next logical step for improvement will be to implement the two-component model directly into the 2-D inversion and to avoid the detour over the Cole-Cole model.

## 8 Conclusions

Broad-band observations of complex electrical resistivity were carried out on a permafrost field site close to the Siberian city Yakutsk. We used the novel complex resistivity measurement device "Chameleon-II", designed for studying electrical dispersion in a wide frequency range in highly resistive regions, especially under cryospheric conditions. The data analysis is

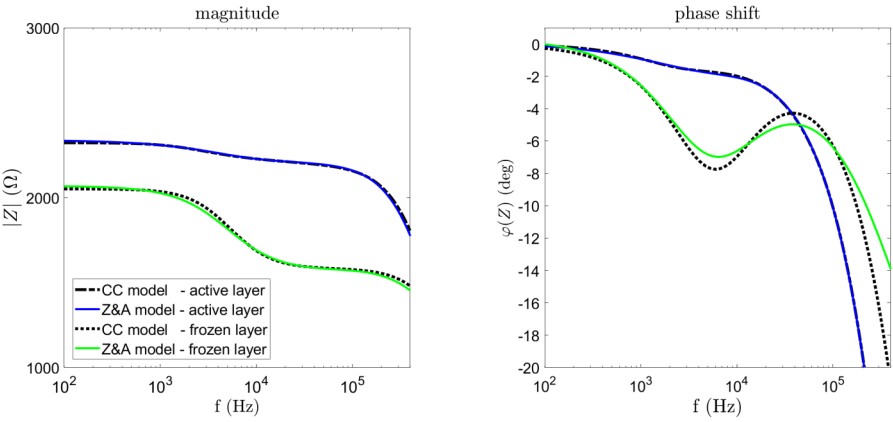

**Figure 12.** Spectra of the magnitude (left) and the phase shift (right) of the impedance $Z$ by the single Cole-Cole model calculated by the AarhusInv 2-D inversion (dashed lines) and the fitted spectra after the model by Zorin and Ageev (2017) (colored lines). The spectra represent exemplarily two different positions within the spacial range of the 2-D inversion, one from the unfrozen active layer (blue) and another within the permafrost layer in around 2 m depth (green), which correspond to the cells outlined in red in fig. 11.

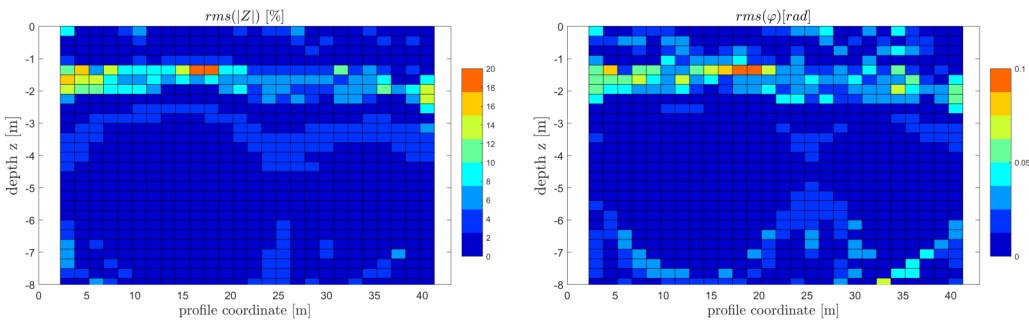

**Figure 13.** Data fit of the ice content estimation in terms of the root mean square for the magnitude $|Z|$ (left panel) and the phase shift $\varphi$ (right panel). The values are displayed for every inversion cell over the 2-D profile.

performed by fitting a permittivity Cole-Cole model to the measured spectra, either for single measurements or within a 2-D inversion, to extract the characteristic frequency dependent behaviour of ice dispersion. Based on the Cole-Cole inversion, we implemented a 2-D estimation of the subsurface ice content using an existing two-component complex resistivity model of frozen soil.

5    The evaluation of the measured HFIP spectra shows a characteristic dispersion in the expected frequency range, which is caused by the known electrical dispersion of ice. A systematic variation of the spectra with the investigation depth is observed, which is reasonably consistent with the structure and measured values from previous borehole and ERT investigations. The unfrozen area within the permafrost body, as well as the shallow active layer, can be distinguished from the areas of frozen ground. We conclude that the HFIP method in combination with our novel procedure to estimate ice content is a useful tool for per-

mafrost research at the field scale. The method is able to separate frozen from unfrozen ground, and more importantly, to provide quantitative estimates of ice content. In the case study discussed here, the estimated ice content values are close to zero where the ground is unfrozen, and larger than zero in frozen sections. The estimated values were compared with ice contents determined from soil moisture data obtained from borehole samples. In the absence of detailed information on lithology and temperature, soil moisture could only be used as an estimate of the upper limit, since there might be fluid water in the soil even below freezing temperature. The comparison reveals that the results obtained from the HFIP method seem to slightly overestimate ice content. A thorough assessment of the achievable accuracy will require more data with more boreholes.

At the local scale, our study confirms the supposed existence of taliks along the profile by verifying that the low resistive zones along the profile indeed have zero ice content. We also conclude that the Chameleon II equipment with its specific electronic concept to avoid EM coupling is feasible and sufficiently robust to be applied over highly resistive ground typically encountered in permafrost research.

The new approach to estimate ground ice content at the field scale using full spectral information, encourages future research. First, it might be conceivable to combine our method and its advantages with existing models, such as the ice content inversion by Wagner et al. (2019) based on the model from Hauck et al. (2011), to further reduce ambiguity inherent in each separate method. Second, the method might be advanced to explore even greater depths. Currently, we are limited by the maximum distance between transmitter and receiver of $50\,\mathrm{m}$, which was considered a maximum reasonable length where distortions by EM coupling could be sufficiently reduced by the specific measuring electronics. The encouraging results obtained here indicate that it could be worthwhile to put further efforts into the development of the measuring system, including a systematic evaluation of data errors and a suitable error model to be used during inversion. In that context, EM induction effects that cannot be avoided by measuring technology, will also play a greater role. These effects increase with the square of the spatial scale, which might require correction or simulation procedures, if greater depths will be investigated.

Further improvements might be achieved by directly implementing the two-component model into the 2-D inversion, thus replacing the Cole-Cole model. The Cole-Cole model appeared well justified by its wide usage to describe impedance spectra of ice-containing material. The choice was also driven by the availability of 2-D invesion code. However, the increasing fitting error with increasing ice content observed in our case study indicates that the Cole-cole model might not be ideal. A novel HFIP 2-D inversion routine that offers more flexibility for parameterization might be developed in future using the PyGIMLi framework (Rücker et al., 2017). Once a flexible parameterization has been achieved, the two-component model we currently rely on Zorin and Ageev (2017), might also be extended or modified to allow the inclusion of more material phases (Bittelli et al., 2004; Stillman and Grimm, 2010).

Considering the broad frequency range over which reasonable data quality can be obtained, the integration of low-frequency polarization mechanisms might be conceivable as well. As illustrated using a tentative implementation of EDL polarization, low-frequency polarization may be neglected for ice content estimation purposes, since the ice relaxation dominates the spectra in the relevant frequency range. In future, the EDL polarization might become an integral part of a broadband model, in order to derive additional parameters, such as CEC or porosity, within a comprehensive theory.

For future studies, the temperature should also be considered in detail as it controls several parameters, such as ice relaxation

time, water conductivity and the ice content itself. Here, we used standard parameters assumed to be valid close to the freezing point. Since detailed temperature data are often not available, the choice of a representative temperature, and the error introduced by inaccurate assumptions, might be critical to understand. Temperature is also important when trying to extract the ice content from soil moisture, as it controls the amount of fluid water present in the soil below freezing temperature. If a temperature profile within a borehole is known, and the ice content can be accurately obtained from frozen cores, it might become feasible to calibrate HFIP results at the borehole location to increase the overall accuracy of spatial ice content distribution estimates.

*Code and data availability.* The HFIP data measured at the field site are available on request to Jan Mudler (j.mudler@tu-bs.be). Access to AarhusInv software can be requested at https://hgg.au.dk/software/aarhusinv/.

*Author contributions.* JM, KB and LL performed the different measurements in the field and analyzed the data. DK and JM developed the code for the ice content estimation. MS supported the process of data analysis. AH supported the whole process of research and wrote several sections of the article. TR developed and built the measurement device and its software. JM prepared the manuscript with contribution from all co-authors.

*Competing interests.* The authors declare that they have no conflict of interest.

*Acknowledgements.* We are grateful to Malte Lührs (TU Braunschweig) for his support of our measurements in Yakutia and Johannes Buckel (TU Braunschweig) for proofreading and support on the manuscript.

The work was sponsored by the German Research Foundation DFG (project HO 1506/22-2). The reported study was partially funded by the Russian Foundation for Basic Research RFBR (projects 20-35-70027 and 20-05-00670).

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
