# Peer review of "Broadband Spectral Induced Polarization for the detection of Permafrost and an approach to ice content estimation – A Case study from Yakutia, Russia"

_The Cryosphere, 2021_

## Author Comment (AC2)

**Authors Response to Review 2 (from 05.08.2021)**

**R**: Referee's comment

**A**: Author's response

**R:** Mudler et al. present a case study using high frequency spectral induced polarization (HFIP) data to detect the frozen/unfrozen layer and estimate the ice content in a permafrost environment. The spectral IP data were fitted using an empirical model to extract the complex dielectric permittivity and DC resistivity parameters. These parameters were interpreted to characterize the frozen/unfrozen layer of the subsurface. The parameters were further used to estimate the ice content. While this manuscript matches the scope of the Cryosphere, it contains a few technical issues in terms of the methodologies and data interpretation.

**Major comments**

**(1)**
Measurement accuracy of HFIP

**R:** This study collected SIP data from 2 Hz to 115 kHz. Particularly, high frequency (HF) data in kHz were mainly discussed as it was stated that polarization of ice occurs in this frequency range. However, the measurement accuracy of HFIP was not evaluated quantitatively. It is well known to the IP community that the four-electrode method results in huge errors at high frequencies. It is very challenging and requires extensive procedures to remove HF errors at the laboratory scale measurements. It is even more difficult to collect high-quality HFIP data at field-scale, especially in a high resistivity environment like this study. This manuscript does discuss the HF error topic (only qualitatively) in Section 4, whereas it does not provide any information concerning instrument accuracy.

**A:** The accuracy of field scale measurements in comparison with lab measurements is a complicated issue, in particular when EM coupling is involved. We tend to disagree with the general statement that it is more difficult at the field scale, as we are not aware of any studies supporting the statement.
The potential problems of data acquisition for HFIP data are caused by the interference from electromagnetic effects. However, the reduction of these effects is taken into account by the measurement device (Chameleon-II), designed to be used in high resistive areas (Radic et al., 2018). Thus, inductive and capacitive cable couplings are minimized by the used hardware. Capacitive coupling between current cables and the subsurface is reduced by integrated cable-earth compensation (CEC) from the instrument side and the software according to Radic & Klitzsch (2012). Inductive coupling effects between current cables and the subsurface can be estimated, for example by the induction number and suggest that over resistive subsurface, such as permafrost, those effects can be neglected in the used range of the measurements.
To give a quantitative estimation of these effects, we will provide and illustrate statistical errors of the measurements for exemplary data with and without the CE compensation. Furthermore, we will provide a simple estimation of the influence of inductive coupling effects.

**(2)**

The models

**R:** I found it difficult to follow the SIP models used in this study. Generally, there are four parameters describing the electrical conduction and displacement/polarization: real conductivity, imaginary conductivity, real permittivity, and imaginary permittivity. It is not clear how these parameters were treated, for example, were the conductivity parameters related to the permittivity parameters? Was any parameter neglected?

**A:** Concerning the choice of permittivity instead of imaginary conductivity, we will add the following explanation in the section below eq. (2):
"In general, there is a choice whether the data interpretation is based on imaginary conductivity, or on the real part of permittivity, because the two are mathematically equivalent. Whereas for low-frequency (<100Hz) SIP measurements, imaginary conductivity is often preferred (Loewer et al. 2017), for high-frequency SIP covering the relaxation of ice, permittivity is generally considered (Bittelli et al., 2004)"

**R:** Specific questions are:
In eq. (1), are $\rho$ and $\varepsilon_r$ complex quantities? If so, is $\varepsilon_r$ the same as $\varepsilon_{r*}$. If not, is $\varepsilon_r$ the same as $\varepsilon_r'$?

**A:** In principle $\rho$ and $\varepsilon_r$ in eq. (1) are real values. We reconsidered eq. (1) and decided that it is not necessary for the following explanations. Therefore, we will take out eq. (1) and instead we will integrate the explanation of the real and imaginary part of $\varepsilon_r$ into eq. (2).

**R:** The description of Eq. (2) is a bit confusing as a Cole-Cole form model does not have the third term. It is reasonable, though, to have the third term to describe the DC conduction, but again the discussion of these parameters is mixed up. It seems that imaginary conductivity was never mentioned, although it is very important for SIP. Besides, as eq. (2) is the key equation for fitting the data, more information is needed to clarify how complex impedance was converted to a complex permittivity.

**A:** The used model (eq. (2)) is the Cole-Cole model with an additional third term that integrates the low-frequency conductivity mechanisms. This model has been previously used for cryospheric investigations by several publications (e.g., Bittelli et al. 2004, Stillman et al. 2010, Grimm & Stillman 2015). Nevertheless, we agree that the naming of this model as "Cole-Cole model" is misleading and we will change and clarify this in the text.
According the imaginary conductivity, see the above Authors Comment.
The conversion from complex impedance to complex permittivity will be clarified in this section, as it is provided e.g. in Przyklenk et al. (2016).

**R:** Eq. (3-5). Ice estimation was made based on these equations. On page 5, line 27 states that three parameters are well known and fixed. These parameters should be provided.
I am also curious how the $\tau_i$ was selected as it is a temperature-dependent parameter. Also, it would be helpful to describe the meaning of parameter k and present and discuss the variations of fitted k.

**A:** We will provide the used literature values of the fixed parameters, named on page 5. The polarization of ice does indeed exhibit a temperature dependence, but is approximated here in the first approach as independent and with the values for temperatures immediately below freezing point, which is valid for the discussed field survey. We will discuss the temperature dependence of the parameters in more detail.

Furthermore, we will illustrate the distribution of fitted parameter k and discuss the meaning of k with regard to Zorin & Ageev (2017).

**(3)**
Data interpretation

**R:** The whole Section 6 describes the raw data from a 1D sounding. However, those data are apparent IP data and do not represent the true electrical responses of the subsurface. Nowadays, these data mostly only serve as a way to assess the raw data quality. Therefore, it is not proper to relate them to a physical process and interpret them so extensively (accounting for half of the results), especially for a non-layered structure as evident from the zones around 'C' in Figure 7.

**A:** We agree that the description of raw data from a 1-D sounding is not common for conventional measurements. Here, they are only discussed because there is little research on the method and its application to ice-bearing subsoils, and we find it useful to illustrate the relationship between the models and the raw data. However, we will shorten this section and make it clearer that these are apparent HFIP data that do not allow for physical or geologic interpretation.

**R:** Besides, there are a few specific questions that need to be addressed:
As the polarization of ice is non-metallic polarization, wouldn't imaginary conductivity be a better parameter to interpret to exclude the effects of variations in water conductivity?

**A:** We follow the typical parameterization describing the ice polarization in the terms of the complex permittivity (see Authors Comment (2)). Possible effects due to water are not explicitly considered at this stage. It would be an approach for further work to use a combined model from the parameterization of several relaxation processes by the conductivity (low-frequent) and permittivity (high-frequency effects), as proposed for example by Loewer et al. (2017). However, since the number of free parameters would increase significantly, and the application and evaluation of HFIP in the field is quite unexplored, the initial approach was to keep the model as simple as possible and focus only on the known ice relaxation. We will present this aspect more clearly in the paper and likewise revisit it in the outlook.

**R:** According to Bittelli et al., 2004, the $\varepsilon_{r,DC}$ of ice is around 100, and $\varepsilon_{r,HF}$ is around 3. Figure 7 shows that $\varepsilon_{r,DC}$ is as high as 600 in the frozen layer even though the ice content is less than 100%. Figure 7 also shows high $\varepsilon_{r,DC}$ values (~200) even in the thawed layer without ice. It may indicate that the applied complex permittivity model is a good choice as the fitted $\varepsilon_{r,DC}$ is too high compared to the theoretical value (e.g., 100 for ice). Figure 7 indicates that thawed layer exhibit large relaxation giving the large difference between $\varepsilon_{r,DC}$ and $\varepsilon_{r,HF}$. This is contradictory to Eq.(4), which states that the permittivity of the ice-free matrix is constant.

**A:** The large values of $\varepsilon r,DC$ are an artefact of the 2-D inversion code. If the data show no relaxation process, some the 5 parameters become poorly constrained, which may lead to unrealistic values. We will discuss and possibly illustrate this issue in the revised version.

**(4)**
**R:** In addition to the above main points, the significance and broad applicability of this manuscript may not be adequate for the Cryosphere journal as the study only has one survey line at one site.

**A:** Since the HFIP method is not widely used and the approach of determining the ice content at the field scale is new, we initially validated it on only one measurement area with some prior knowledge on ice content. It is logistically quite expansive to gain access to sites where external information on ice content is available and the method can be validated. We hope to generate some interest in our method which will initiate future fieldwork and more case histories.

**Some suggestions**

**A:** The authors will adjust all suggestions noted by the reviewer.

**Literature**

M. Bittelli, M. Flury, K. Roth: Use of dielectric spectroscopy to estimate ice content in frozen porous media, Water Resources Research, Vol. 40, W04212, 2004

R.E. Grimm and D.E. Stillman: Field Test of Detection and Characterisation of Subsurface Ice using Broadband Spectral-Induced Polarisation, Permafrost and Periglacial Processes, Vol. 26, 28-38, 2015

M. Loewer, T. Günther, J. Igel, S. Kruschwitz, T. Martin, N. Wagner: Ultra-broad-band electrical spectroscopy of soil and sediments – a combined permittivity and conductivity model, Geophysical Journal International, Vol. 210, 1360-1373, 2017

T. Radic and N. Klitzsch: Compensation technique to minimize capacitive cable coupling effects in multi-channel IP systems, Near Surface 2012, Paris, France, 2012

T. Radic, A. Hoerdt, J. Mudler: CHAMELEON II - Field Equipment for Resistivity Measurements up to 230 kHz, 5[th] International Workshop on IP, Newark, USA, 2018
D.E. Stillman, R.E. Grimm, S.F. Dec: Low-Frequency Electrical Properties of Ice-Silicate Mixtures, Journal of Physical Chemistry B, Vol. 114, 6065-6073, 2010

N. Zorin and D. Ageev: Electrical properties of two-component mixtures and their application to high-frequency IP exploration of permafrost, Near Surface Geophysics, Vol. 15, 603-613, 2017

---

## Author Response (AR1)

**Authors Response to Review 1 (from 22.07.2021)**

**R**: Referee's comment

**A**: Author's response

**General Comments**

**R:** The paper by Mudler et al. is devoted to the problem of ice content estimation in the buried rocks using non-invasive geophysical techniques, namely, the high-frequency induced polarization (HFIP) method. The manuscript is well-structured, rather concise and supported with sufficient illustrative materials. It contains novel interesting results and clearly deserves publishing even without any significant corrections. However, below there are some comments and suggestions, which I believe may help the authors to further improve the scientific quality of the paper.

**Specific comments**

**(1)**
**R:** The main idea of the manuscript is to estimate the ice content in the buried rocks by using the two-component weighted power mean (WPM) model for fitting measured broadband HFIP spectra. For this purpose the authors first invert the observed 2-D HFIP data by means of the conventional Cole-Cole model and then make separate additional inversions of the revealed Cole-Cole response within each model cell by means of the two-component WPM formula. This approach is vulnerable to criticism, since the WPM and Cole-Cole functions could not be generally converted to each other, hence application of the Cole-Cole inversion to the data complying with the WPM model may only introduce additional errors and thus appears to be rather undesirable. If the authors believe that the WPM model is the best choice for quantitative description of the observed HFIP data, then the most natural way of handling them would be direct 2-D inversion for the WPM model parameters, without unnecessary intermediate use of the Cole-Cole function. Consider trying this approach if there are technical capabilities to do so – it may probably yield better results, provided that the non-ice IP effects in rocks are relatively small (otherwise some combination of the Cole-Cole conductivity + WPM permittivity models could be required instead).

**A:** We agree that the approach of first using the 2-D Cole-Cole fit and afterwards adjust the WPM model appears cumbersome and needs justification.
The reason why this approach is taken is that all available possibilities to extract the two-dimensional distribution of the total spectral information from HFIP data are based on some type of Cole-Cole parameterization. The code AarhusInv used here (Mudler et al., 2019), is not freely modifiable. Alternatively, the code based on the pyGIMLi framework (Günther and Martin, 2016) might be considered, but so far, only the single Cole-Cole model has been implemented. Nevertheless, implementing the WPM directly into the 2-D inversion is exactly our goal for the future.
We added a discussion of this aspect in the results and conclusions section, and we tried to make clearer in the document why it is currently necessary to use the Cole-Cole model first. We understand our approach as a first step of a two-dimensional ice content determination from field data from the single method of HFIP, which does not exist yet.

**(2)**

**R:** The choice itself of the employed model and its variable parameters should be discussed in more detail, if possible. For quantitative description of the HFIP response of an ice-bearing rock one may use the 2-component (Zorin, Ageev, 2017), 3-component (Stillman et al., 2010) or 4-component (Bittelli et al., 2004) WPM formula, not to mention the other potentially applicable mixing models, such as that of Hanai and Bruggeman. Why did you choose to employ the 2-component WPM for your data set? Are the temperature and clay content in the frozen layer under study low enough to consider all non-ice sources of IP effect negligible? Is it legit to fix the relaxation time constant of ice as a known value (page 5 lines 27-29), while it could in general vary by several times depending on ice purity and temperature?

**A:** We use the 2-component model due to its simplicity so that the number of free parameters is initially kept as small as possible. Nevertheless, we agree that one problem is that low-frequency IP effects are not adequately taken into account.
However, since we apply the model to the simple Cole-Cole fit, i.e., with one relaxation, the coexistence of the ice relaxation and IP relaxation of a data set is not foreseen at this stage. Nevertheless, extending this to a 3-component model (extended by water) could lead to better reproduction of effects in unfrozen regions (unfrozen active layer, talik). The intention to increase the model to 3 or 4 components and thus to consider IP effects, as well as the temperature and other aspects of the subsurface, is the goal of an upcoming project. However, at this stage and as a proof of concept, we consider the presented approach as reasonable. We discuss this aspect in a little more detail in the conclusions of the revised version.

**R:** To answer these and other related questions it should be useful to provide the inversion results for all parameters of the employed WPM model and discuss more thoroughly the quality of data fitting, especially within the ice-bearing cells: the reported average misfit of 20% for amplitude and 0.15 rad = 8.6 degrees for phase (page 16 lines 7-8) appears to be rather high, but there are no illustrations showing how exactly and at which frequencies the actual data diverge from the best-fit model, so it is difficult to understand how the employed model should be modified to achieve better results.

**A:** We agree that the data fitting should be discussed and presented in more detail. Therefore, we added a section where we present and discuss the fitting of individual spectra in the range of ice-bearing and active layer. Furthermore, during the review process, we revised the procedure of ice content estimation and we are now able to achieve a better fit by imposing limits of the model parameters within the fitting process. The average fitting error for both amplitude and phase shift in the area of the ice-bearing subsurface is now considerably smaller in the revised version. Note that the results of ice content did not change significantly by the modification. We also added a discussion on the sources of the remaining errors and a potential future solution.
We also agree that it would be useful to illustrate the distribution of additional model parameters. Since all parameters would be too much, we focus on presenting and discussing the distribution of model parameter k.

**Technical corrections**

**A:** The authors adjusted the technical corrections noted by the reviewer.
Only the remark about linear scaling in fig. 5 of the revised version (fig. 4 in the originally submitted version) has been retained, because the representation of the additional borehole data from the Shestakovka field site by Lebedeva et al. (2019) was chosen identically and thus a comparability should be given.

**Authors Response to Review 2 (from 05.08.2021)**

**R**: Referee's comment

**A**: Author's response

**R:** Mudler et al. present a case study using high frequency spectral induced polarization (HFIP) data to detect the frozen/unfrozen layer and estimate the ice content in a permafrost environment. The spectral IP data were fitted using an empirical model to extract the complex dielectric permittivity and DC resistivity parameters. These parameters were interpreted to characterize the frozen/unfrozen layer of the subsurface. The parameters were further used to estimate the ice content. While this manuscript matches the scope of the Cryosphere, it contains a few technical issues in terms of the methodologies and data interpretation.

**Major comments**

**(1)**
Measurement accuracy of HFIP

**R:** This study collected SIP data from 2 Hz to 115 kHz. Particularly, high frequency (HF) data in kHz were mainly discussed as it was stated that polarization of ice occurs in this frequency range. However, the measurement accuracy of HFIP was not evaluated quantitatively. It is well known to the IP community that the four-electrode method results in huge errors at high frequencies. It is very challenging and requires extensive procedures to remove HF errors at the laboratory scale measurements. It is even more difficult to collect high-quality HFIP data at field-scale, especially in a high resistivity environment like this study. This manuscript does discuss the HF error topic (only qualitatively) in Section 4, whereas it does not provide any information concerning instrument accuracy.

**A:** The accuracy of field scale measurements in comparison with lab measurements is a complicated issue, in particular when EM coupling is involved. We tend to disagree with the general statement that it is more difficult at the field scale, as we are not aware of any studies supporting the statement.
However, we agree that errors are an issue for HFIP measurements, in particular at high frequencies. Therefore, we significantly extended the discussion of potential errors in the measurement. We also included a new figure that helps to evaluate the errors.
One panel of the new figure deals with errors caused by capacitive coupling between transmitter cable and the ground (CE coupling) and shows data with and without the CE compensation. The other panel shows an example of reciprocal measurements with the Chameleon-II device, which are generally considered a good measure of data error (e.g. Flores Orozco et al., 2012). In reciprocal measurements, the roles of transmitter and receiver are interchanged, which should theoretically provide identical results if no systematic errors are present. We also provide quantitative error estimates in the revised version.

**(2)**
The models

**R:** I found it difficult to follow the SIP models used in this study. Generally, there are four

parameters describing the electrical conduction and displacement/polarization: real conductivity, imaginary conductivity, real permittivity, and imaginary permittivity. It is not clear how these parameters were treated, for example, were the conductivity parameters related to the permittivity parameters? Was any parameter neglected?

**A:** Concerning the choice of permittivity instead of imaginary conductivity, we added the following explanation in the section below eq. (2):
"In general, there is a choice whether the data interpretation is based on imaginary conductivity, or on the real part of permittivity, because the two are mathematically equivalent. Whereas for low-frequency (<100Hz) SIP measurements, imaginary conductivity is often preferred (Loewer et al. 2017), for high-frequency SIP covering the relaxation of ice, permittivity is generally considered (Bittelli et al., 2004)"

**R:** Specific questions are:
In eq. (1), are $\rho$ and $\varepsilon_r$ complex quantities? If so, is $\varepsilon_r$ the same as $\varepsilon_{r*}$. If not, is $\varepsilon_r$ the same as $\varepsilon_r'$?

**A:** In principle $\rho$ and $\varepsilon_r$ in eq. (1) are real values. We reconsidered eq. (1) and decided that it is not necessary for the following explanations. Therefore, we removed eq. (1) and integrated the explanation of the real and imaginary part of $\varepsilon_r$ into eq. (2).

**R:** The description of Eq. (2) is a bit confusing as a Cole-Cole form model does not have the third term. It is reasonable, though, to have the third term to describe the DC conduction, but again the discussion of these parameters is mixed up. It seems that imaginary conductivity was never mentioned, although it is very important for SIP. Besides, as eq. (2) is the key equation for fitting the data, more information is needed to clarify how complex impedance was converted to a complex permittivity.

**A:** The used model (eq. (2)) is the Cole-Cole model with an additional third term that integrates the low-frequency conductivity mechanisms. This model has been previously used for cryospheric investigations by several publications (e.g., Bittelli et al. 2004, Stillman et al. 2010, Grimm & Stillman 2015). Nevertheless, we agree that the naming of this model as "Cole-Cole model" is misleading and we clarified this in the revised text. Concerning the imaginary conductivity, see the above Authors Comment. The conversion from complex impedance to complex permittivity has been modified in this section, we hope it is clear now.

**R:** Eq. (3-5). Ice estimation was made based on these equations. On page 5, line 27 states that three parameters are well known and fixed. These parameters should be provided.
I am also curious how the $\tau_i$ was selected as it is a temperature-dependent parameter. Also, it would be helpful to describe the meaning of parameter k and present and discuss the variations of fitted k.

**A:** In the revised version, we now provide the used literature values of the fixed parameters on page 5.
The polarization of ice does indeed exhibit a temperature dependence, but is approximated here in the first approach as independent and with the values for temperatures immediately below freezing point, which is valid for the discussed field survey. We discuss the temperature dependence of the parameters in more detail. Furthermore, we added a section and a figure in which we illustrate the distribution of the fitted parameter k and discuss the meaning of k with regard to Zorin & Ageev (2017).

**(3)**
Data interpretation

**R:** The whole Section 6 describes the raw data from a 1D sounding. However, those data are apparent IP data and do not represent the true electrical responses of the subsurface. Nowadays, these data mostly only serve as a way to assess the raw data quality. Therefore, it is not proper to relate them to a physical process and interpret them so extensively (accounting for half of the results), especially for a non-layered structure as evident from the zones around 'C' in Figure 7.

**A:** We agree that the description of raw data from a 1-D sounding is not common for conventional measurements. Here, they are only discussed because there is little research on the method and its application to ice-bearing subsoils, and we find it useful to illustrate the relationship between the models and the raw data. Therefore, we would like to refrain from shortening this section too much and hope that this is acceptable to the reviewer. However, we will make it clearer that these are apparent HFIP data that do not allow for physical or geologic interpretation.

**R:** Besides, there are a few specific questions that need to be addressed:
As the polarization of ice is non-metallic polarization, wouldn't imaginary conductivity be a better parameter to interpret to exclude the effects of variations in water conductivity?

**A:** We follow the typical parameterization describing the ice polarization in the terms of the complex permittivity (see Authors Comment (2)). Possible effects due to water are not explicitly considered at this stage. It would be an approach for further work to use a combined model from the parameterization of several relaxation processes by the conductivity (low-frequent) and permittivity (high-frequency effects), as proposed for example by Loewer et al. (2017). However, since the number of free parameters would increase significantly, and the application and evaluation of HFIP in the field is quite unexplored, the initial approach was to keep the model as simple as possible and focus only on the known ice relaxation. We presented this aspect more clearly in the paper and likewise revisit it in the outlook.

**R:** According to Bittelli et al., 2004, the $\varepsilon_{r,DC}$ of ice is around 100, and $\varepsilon_{r,HF}$ is around 3. Figure 7 shows that $\varepsilon_{r,DC}$ is as high as 600 in the frozen layer even though the ice content is less than 100%. Figure 7 also shows high $\varepsilon_{r,DC}$ values (~200) even in the thawed layer without ice. It may indicate that the applied complex permittivity model is a good choice as the fitted $\varepsilon_{r,DC}$ is too high compared to the theoretical value (e.g., 100 for ice). Figure 7 indicates that thawed layer exhibit large relaxation giving the large difference between $\varepsilon_{r,DC}$ and $\varepsilon_{r,HF}$. This is contradictory to eq.(4), which states that the permittivity of the ice-free matrix is constant.

**A:** The large values of $\varepsilon_{r,DC}$ are an artefact of the 2-D inversion code, due to the fact that the procedure assumes a Cole-Cole relaxation model for all areas of the subsurface. If the data show no relaxation process, some the 5 parameters become poorly constrained, which may lead to unrealistic values. Nevertheless, the data can also be fitted in the areas assumed to be ice-free, but the interpretation there should be focused on the reliable parameters, excluding $\varepsilon_{r,DC}$ and relaxation time. We added a brief discussion of this issue in the revised version.

**(4)**
**R:** In addition to the above main points, the significance and broad applicability of this manuscript may not be adequate for the Cryosphere journal as the study only has one survey line at one site.

**A:** Since the HFIP method is not widely used and the approach of determining the ice content at the field scale is new, we initially validated it on only one measurement area with some prior knowledge on ice content. It is logistically quite expansive to gain access to sites where external information on ice content is available and the method can be validated. We hope to generate some interest in our method which will initiate future fieldwork and more case histories.

**Remarks under the heading "Some suggestions"**

**A:** We implemented the suggestions noted by the reviewer.

**Literature**

M. Bittelli, M. Flury, K. Roth: Use of dielectric spectroscopy to estimate ice content in frozen porous media, Water Resources Research, Vol. 40, W04212, 2004

A. Flores Orozco, A. Kemna and E. Zimmermann: Data error quantification in spectral induced polarization imaging, Geophysics, Vol. 77, E227-E237, 2012

R.E. Grimm and D.E. Stillman: Field Test of Detection and Characterisation of Subsurface Ice using Broadband Spectral-Induced Polarisation, Permafrost and Periglacial Processes, Vol. 26, 28-38, 2015

M. Loewer, T. Günther, J. Igel, S. Kruschwitz, T. Martin, N. Wagner: Ultra-broad-band electrical spectroscopy of soil and sediments – a combined permittivity and conductivity model, Geophysical Journal International, Vol. 210, 1360-1373, 2017

T. Radic and N. Klitzsch: Compensation technique to minimize capacitive cable coupling effects in multi-channel IP systems, Near Surface 2012, Paris, France, 2012

T. Radic, A. Hoerdt, J. Mudler: CHAMELEON II - Field Equipment for Resistivity Measurements up to 230 kHz, 5th International Workshop on IP, Newark, USA, 2018
D.E. Stillman, R.E. Grimm, S.F. Dec: Low-Frequency Electrical Properties of Ice-Silicate Mixtures, Journal of Physical Chemistry B, Vol. 114, 6065-6073, 2010

N. Zorin and D. Ageev: Electrical properties of two-component mixtures and their application to high-frequency IP exploration of permafrost, Near Surface Geophysics, Vol. 15, 603-613, 2017

---

## Author Response (AR2)

**Authors Response to Report #1 (from 02.05.2022)**

**General Comments**

the subject of the article is well adapted to a publication in cryosphere. The estimation of ice content from geophysical measurements is undoubtedly a very important purpose but complex and difficult subject to solve!

The complexity of the problem comes from the physic of the phase change but also from the methods used. The article is well written and the illustrations are good.

It is a pity that the authors did not validate the model with your synthetic cases or with laboratory measurements beforehand. the study site is interesting for the complexity in the distribution of permafrost.

thanks to the authors for this paper.

**Reply:** We appreciate the generally positive comments. We are aware that we are dealing with a complex problem and we tried to improve the material along the suggested lines.

Specific comments

**1) plan and purpose**

the distribution of the parts is to be improved. the model used is too little developed whereas the instrumental part is too present.

The development of new instruments and measurement techniques in the field is a very interactive and an important subject but I advise to explain it another paper.

Indeed, the instrumental problems that are mentioned are numerous and are covered too quickly. Moreover, this is not really the subject of the article which is the estimation of ice content. if I am not mistaken.

The presentation of the results is a bit long, especially in relation with the geological context. This long presentation of the results is done at the expense of the discussion which is not sufficiently developed, especially on the chosen hypotheses and on the model.

**Reply**

The intention of the paper was to outline the entire procedure to estimate ice content at the field scale. We believe that the succesful application at the field scale - as compared to work in the laboratory - is a major advancement. Therefore, we would prefer to keep the instrumental section in the paper; in particular, since some of the material presented here came in as a result of reviewer's comments on the previous version.

We agree, however, that the model might be better discussed and justified, and therefore, we shortened the instrumental section and expanded the theoretical section and the discussion of the model.

**2) Bibliography and freezing problem**

the theoretical part is to be developed. It is focused on one type of approach without mentioning or discussing other models. Therefore, there is a lack of bibliographic references, especially on other approaches to model the SIP or HSIP response of an icy environment. the focus of the model only on the polarization of the ice leads to a too simplistic model to model correctly the SIP response of an icy medium. at least, you have to justify much more the assumptions.

**Reply**

We extended the theoretical section, and we now include more references, in particular on low-frequency polarization. Alternative high-frequency models are already mentioned in the theoretical section, and we are not sure which ones are missing.
In order to better justify our assumptions, we tentatively expanded our model to add low-frequency-polarization in an icy environmend using a model published in literature. Using this extended model, we carry out a small synthetic study to show that the low-frequency polarization may indeed be neglected.
We agree that our model might still be extended in future in order to include more processes, and we extend the discussion in the conclusons section to cover this aspect.

**Comment**
in the introduction of the article it is mentioned that it is necessary to develop a tool to follow the evolution of permafrost melting (or its renewal). a presentation of the thermal process is necessary to add. I am thinking in particular of the frost curve to be presented (e.g. Watanabe and Mizoguchi 2002). It is important to remember that permafrost consists of a solid fraction, a pore space filled with air, water and ice. Depending on the temperature and the initial water content of the soil, the ice content of a soil can be low. I'm not sure the assumption of just focusing on ice polarization is acceptable or shrewd. moreover, along an ERT profile, the lateral variations of lithology necessarily cause a variation in the importance of the polarizations of the medium and of the ice.

**Reply**
Actually, we are not sure which section in the introduction the reviewer is referring to, as we believe we are not using the word "evolution" in our introduction and do not specifically expand on permafrost melting and renewal anywhere. However, we agree that the aspect of the frost curve, and the fact that ice content may strongly vary close to 0°C is important, in particular in the context of the temperature dependence of the ice polarization. We thank the reviewer for pointing out the corresponding literature. Therefore, we included a paragraph in the corresponding section, and expand the corresponding discussion in the conclusions. We prefer, however, not to include another figure, because this is not quite along the main lines of our present work, and would further expand the material.

**3) Model**
from the outset, the model used is intended to be high frequency in order to focus on the polarization of the ice. however, there is no mention of the physical processes explaining why there is high frequency ice polarization (Bjerrum effect, interface polarization...). by the way, what is the frequency range of this model? we are talking about high frequency (greater than 100khz) for the ice effect but also at much lower frequencies of the order of 100 Hz. In this case, for low frequencies, the broadband model must integrate the polarizations of Maxwell

Wagner and of the electric double layer. otherwise, one must clearly justify why the polarization of the electric double layer is negligible.

**Reply**
We add a section to discuss the frequency range of validity and low-frequency polarization. We also add a figure with a synthetic study where we tentatively include polarization of the electric double layer to justify our decision to neglect it. We also point out that Maxwell Wagner polarization is implicitly included in our model (the Zorin and Ageev model), and add a reference on the ice polarization. Nevertheless, we are aware that our model may still be expanded in future to include more processes, and we add a corresponding section in the conclusions.

**Comment**
the addition of a figure from synthetic data showing the impact of the variation of a parameter on the overall response will be appreciated. Finally, the development of a broadband model is very complex even for a soil without frost and requires validating this model from laboratory measurements. Validation of the model remains to be done. the site studied should rather be seen as an application of this model.

**Reply**
We follow the reviewer's suggestion to add a figure showing the impact of a parameter variation. In order not to blow up the material too much, we merge with figure with the one used to illustrate the effect of low frequency polarization.
Concerning validation, Zorin and Ageev (2017) did test their model in the laboratory, but we agree that further validation will be useful and now mention this in the paper.

**4) Instrumental part**
the instrumental part is very interesting, but it could be reduced or make a paper in itself. Adding some advice, benchmarking on acquiring good measurements will be appreciated.

**Reply**
We are not sure whether the instrumental part would make a paper in itself, as we describe advancements over a predecessor equipment that has already been published. On the other hand, as this is the first published application of this piece of equipment, we believe that a section on instrumentation is necessary. Nevertheless, we shortened the instrumental section where feasible and added some advice on the usage.

**5) Results**
the presentation part of the site is well built as well as the result part. it would be appreciated to improve legend and caption of the figures in relation to the result (indicate the limits of different area, the active layer, the permafrost..). The result part could be condensed especially in the description of the different areas more or less frozen in order to add a few lines on the processing of the measurements (number of measurements acquired and filtered, frequencies used…) and on the inversion.
what is missing is a more relevant discussion, particularly on the model and the assumptions used.

**Reply**
We improved figures and captions and condensed the results part. We also extended the

discussion of the model, but actually not in the results section, but in the model description section.

**Comments**

Abstract:
L4-5: "The High-Frequency Induced Polarization method (HFIP) enables the measurement of the frequency dependent electrical signal of the subsurface"

Could you specify the frequency range?

**Reply**
Done

**Comment**
The terms "electrical signal" it is not the most suitable. Maybe specify the name of the parameters that depend on frequency (conductivity and polarization) or the name of the processes (polarization and conduction processes).

L 6: "In contrast to the well-established Electrical Resistivity Tomography (ERT), the usage of the full spectral information provides additional physical parameters of the ground"

additional physical parameters: could you add some examples CEC, clay content, ice content...

**Reply**
We reformulated that section accordingly

**Comment**

L 11-14: "Amongst other improvements, compared to a previous generation, the new system is equipped with longer cables and larger power, such that we can now achieve larger penetration depths up to 10m."

The paper is not about the development of a new measuring instrument but rather about a methodology to determine the ice content. Maybe reduce the paragraph on this new instrument and detail the methodology in the abstract.

**Reply:**
As mentioned above, we also consider the instrumental part and the case history important. Nevertheless, we slightly compacted the instrumental section in the abstract and inserted a sentence on the model.

**Comment Page 2**
L 5: "The frequency dependent electrical properties of ice have been studied by several authors over the past decades in the laboratory for pure ice as well as for ice within sediment mixtures (e.g. Auty and Cole, 1952; Hippel, 1988; Bittelli et al., 2004; Grimm et al., 2015; Artemov, 2019).
complete some references for example:
Coperey, A., A. Revil, et al. « Low-Frequency Induced Polarization of Porous Media

Undergoing Freezing: Preliminary Observations and Modeling ». Journal of Geophysical Research: Solid Earth, https://doi.org/10.1029/2018JB017015.
Olhoeft, G. R. (1977). Electrical properties of natural clay permafrost. Canadian Journal of Earth Sciences, 14(1), 16–24. https://doi.org/10.1139/e77-002

**Reply**
We include the two suggested references. A complete list would probably be subject of a review paper, and we would like to keep the list limited to the paper we considered specifically relevant for us.

**Comment**
L 19: "Attempts have been made to estimate ice content with one method only" complete some references.
maybe you think about studies like Duvillard, 2018 « Three-Dimensional Electrical Conductivity and Induced Polarization Tomography of a Rock Glacier ». Journal of Geophysical Research: Solid Earth.

**Reply**
Our formulation was misleading, as we were referring only to the references already included. We reformulated the corresponding section. Duvillard et al (2018) do not make an attempt at quantitative ice content estimation, and therefore we do not include the paper here.

**Comment**
L 20: "A promising parameter is the frequency-dependent electrical permittivity."
there is also the work of Petrenko, 1993; Petrenko and Ryzhkin, 1997, which is very interesting on the dielectric properties of ice.
**Reply**
We included the references, but at an earlier section in the introduction.

**Page 3 Comment**

L 6: "The previous studies were limited to qualitative interpretation with respect to ice content. "
Please add some references

**Reply**
We reformulated the section. The statement refers to the studies that have been discussed in the previous paragraph, and not to additional literature.

**Comment**
L 6: Partly due to the lack of penetration depth of the acquisition system, […]
please, explain why?
I do not see why the lack of depth of investigation causes a quantitative interpretation of the ice content. Moreover, the depth of investigation in a resistant medium is not so bad compared to a conductive medium which requires more injection power and larger profile. Can you detail and give examples of study. thank you

**Reply**
Apparently, our formulation was misleading. Indeed, the penetration depth is not related to the ice content estimation itself. We reformulated the corresponding section to avoid misunderstandings.

**Comment**

L 19: "It contains the full information about the two material dependent properties of the ground: the electrical resistivity _ and the relative dielectric permittivity "

general remark not requiring change: it is better to speak or express electrical conductivity rather than electrical resistivity in the purely physical or petrophysical sense, i.e., electrical conductivity is really a material property (in the same way as thermal conductivity). Talking in electrical resistivity is right but less relevant from a physical point of view.

**Reply**

We appreciate the remark, but since the terminology is an ever-ongoing issue which we cannot solve here, we indeed would like to leave the formulation as it is.

**Comment**

L 23: "Several reasons for polarizability are known, which can be distinguished by their strength and their occurrence in frequency range (Loewer et al., 2017)."

the terms "process" is more correct instead of reasons.

**Reply**

we changed it.

**Page 4 Comment**

L 8: "In general, there is a choice whether the data interpretation is based on imaginary conductivity, or on the real part of permittivity, because the two are mathematically equivalent. Whereas for low-frequency (< 100Hz) SIP measurements, imaginary 10 conductivity is often preferred (Loewer et al., 2017), for high-frequency SIP covering the relaxation of ice, permittivity is generally considered (Bittelli et al., 2004)."

Yes, in fact for broad band SIP, this is better to consider the couple (conductivity and the permittivity). Each parameter is related to a specific process. You can find some extra information with Australian group papers like N. Wagner, T. Bore, A. Scheuermann…

**Reply**

We studied some of the papers by that group, but we are not sure which ones you are referring to. We find that our justification of using both conductivity and permittivity is compact and sufficient, and would prefer not to add more references without specific need.

**Comment**

L 26: "The inversion leads to the distribution of all five model parameters, […]"

What are these five parameters?

**Reply**

These are the same five Cole-Cole parameters defined previously around eq. (1), We added a sentence to clarify this

**Comment**

L 32: "and on physical models, such as the Maxwell-Wagner polarization, (e.g. Kozhevnikov

and Antonov, 2012; Zorin and Ageev, 2017).”
What do you think of approaches that are based on the polarization of the electric double layer i.e. that do not take into account the polarization of the ice (in the strict sense) but its impact on the polarization of the double layer. (see for example coperey 2018, coperey 2021). A broad brand model based only on the polarization of the ice seems insufficient to me, especially if it wants to be broad band model.

**Reply**
For the purpose of a synthetic modelling study, we now included the Coperey et al., 2018 model and included a new figure to discuss the effect of low-frequency polarization. We discuss why (at the moment) this is not included into the inversion procedure.

**Comment Page 5**
L 5: “In that theory it is assumed, that the polarization is fully caused by the ice fraction.”
It depends on the frequencies you consider i.e. at 100 Hz you have the beginning of the polarization of the Maxwell Wagner and possibly the end of the polarization of the electric double layer for fine minerals like clay which are also very polarizable.

**Reply**
See above; we discuss this aspect in more detail in the new section. Note that Maxwell Wagner polarization is implicitly included in our model, which we now mention in an earlier section.

**Comment Page 6**
L10: “[…] the temperature dependence of the electrical parameters has been neglected.”
the variation of the resistivity is 2%/°C. it is not because you are close to the melting point that you can neglect this dependence i.e. which is also valid below the melting point. Indeed, the dependence is linked to the mobility of the ions and below the melting/freezing point, there is still some liquid water in the medium.
You can neglect this dependence if the system you are studying does not vary in temperature (or very little) or if you are studying at a given time.

**Reply**
We modified and expanded the section on temperature dependence, including a discussion of a possible temperature change with time.

**Comment L10:** “Furthermore, other factors such as the clay content and, in general, low-frequency polarization effects were neglected, since the resulting effects in materials containing ice are much smaller than the polarization of ice.”
Please explain. This hypothesis must be really justified and clearly indicated in which case it is valid (e.g. low CEC or clay content, low salinity, frequency range used).
Precisely, neglecting the polarization of the EDL in some cases can lead to a wrong estimation.

**Reply**
In order to better justify the model, we added a figure with a simulation study where EDL polarization is included. We conclude from the figure that our assumption is justified and discuss under which conditions the assumption may fail. At the same time, we also try to keep the discussion compact in order not to excessively blow up the material. We are aware that our model is the first stage of a possibly long development, and it may not be feasible at this stage to comprehensively discuss all aspects.

**Comment Page 6**
L23: "Impedance measurements at the field scale with 4-point configurations up to 230kHz pose special challenges on the hardware."

why this specific frequency?
Is the 4-point measurement still judicious to use at high frequency?

**Reply:**
The specific frequency is actually irrelevant in this context; it is given by the specific hardware. We rewrote the section to make clear that we need frequencies > approx. 100 kHz. We are not sure about the second comment, as no possible alternative is known to us. We added a line and a reference to explain that 2-point measurements would not be feasible.

**Comment Page 12**
L9: "The phase shift is shown up to 115kHz, however, for the magnitude, values are only displayed until 60kHz, […]"
if from 60 kHz, the magnitude values have too much error then how can we be sure that the phase shift values are good?
moreover, for the models to be used it is necessary to have the magnitude/phase shift couple. What is then the interest to show the phase shift beyond 60 kHz ?

**Reply**
The determination of the phase shift and the magnitude rely on different parameters (the phase shift mainly on timing, the magnitude on an absolute impedance) and thus it is possible to obtain different errors and data quality for both.
The interest in showing both lies in the illustration of the spectral behaviour that can be well matched by the theory.
We slightly reformulated the section to consider the comment.

**Comment**
L18-19: "If we assume electrical parameters resistivity and permittivity as independent of frequency, […]"
why do you want to make these parameters independent of frequency, if this is the case why do measurements in the spectral domain?

**Reply**
If there is no ice, the parameters become independent of frequency. The section is meant to show that these specific spectra can be explained without ice. One would still make the measurements in spectral domain because otherwise one would not know that there is no ice. We reformulated the section accordingly.

**Comment Page 14**
L30:" The resistivity (panel a) is the same parameter as determined by other electric or electromagnetic methods in geophysics, for example by ERT, and can be compared with those results."

not really, the different methods excite the medium in different ways (assumptions,

frequencies, processes, volume of soil investigated are different). The profiles will therefore be different. What do you think about it?

**Reply**
If all methods do what they claim, i.e. determine a resistivity value at a certain location in space, these values should be the same. Of course, the different ways to obtain the value, including those named by the reviewer, but also the choice of inversion, lead to differences. We reformulated the sentence, and added a sentence further below when figure 9 is being discussed, but we do not want to go too deeply into this discussion, as it is slightly off-topic and refers to any comparison between DC resistivities obtained with different methods.

**Comment Page 15**
L10: "The relaxation time for this layer is in the range of literature value for ice relaxation close to the melting point."

Please add a value range and references.

**Reply:**
We added a range and references.

**Comment Page 16**
L19-20: "The borehole information on frozen and unfrozen state, in combination with the soil moisture measurements (fig. 5) even allows a quantitative assessment of the estimated ice content. The soil moisture in the frozen sections of the borehole can be directly transferred to ice content."

even when the soil is frozen, there is still a liquid fraction present in the soil. thus, the water content cannot be converted directly into ice content. it is necessary to know the freezing curve (see Watanabe 2005). especially when you are in the active zone

one could possibly admit a direct relation between water content and ice content under the eutectic of water (approximately -21°C) and still, there is the segregation of the salts present in water.

**Reply**
We are not sure which Watanabe (2005) publication the reviewer is referring to, as we could not find a publication with that particular reference. We assume that actually Watanabe and Mizoguchi (2002) was meant. We agree that the amount of unfrozen water content below freezing is relevant, and added a discussion of this aspect. We now better distinguish between ice content and water content, and also slightly modified our conclusions.
However, as this aspect concerns rather the estimation of ice content from frozen cores rather than the HFIP method itself, we prefer not to go too deeply into this subject, and did not add a figure, for example.

**Comment Figure 2:**
How do you explain the few percentage differences, especially on the amplitude for the

reciprocal measurements? Why did you choose measurements from an alpine site and not from the Yakutia site?

**Reply**
The dipole-dipole configuration generally does not have the largest signal strength, and also the conditions were relatively harsh at that site.
As mentioned in the paper, we did not measure reciprocals at during the Yakutia survey, and therefore the figure is considered a general performance indicator rather than a specific evaluation of the error. We modified the corresponding sections accordingly.

**Comment Figure 5:**
what do the colours represent?
**Reply**
We are not sure what colours the question refers to. the colours in the lithology bare are described in the figure caption, so we do not see what we could change.

**Comment figure 6:**
the high frequency adjustment is only controlled by the phase?
maybe you can change the colour of the high frequency fit curve for the |z| magnitude and explain its behaviour in the figure legend.
**Reply**
Yes, the phase shift controls the high-frequency adjustment. We added a sentence in the main text, and added some information in the caption, but we would prefer to leave the figure itself unchanged.

**Comment Figure 7:**
the text next to the peak is not very relevant (increasing/decreasing peak with depth), it is better to indicate the relationship between the ice content and the peak size

**Reply**
We changed the figure

**Comment**
figure 8: Label the different sections in terms of zone (active zone, frozen zone).

**Reply**
The figure is initially intended to show only the parameters of the inversion. A differentiated interpretation with respect to the layers is given in detail in the text. Since the active zone (max. 50 cm) covers a very small area of the individual sections in the figure and the layers show significant horizontal variations (Permafrost / Talik), we consider an appropriate labeling to be difficult. Therefore, we prefer to leave the figure unchanged.

---

## Author Response (AR3)

**Authors Response to the accepted version from 12.10.2022**

We thank the editor for accepting the manuscript.

As requested by the editor, we have revised the references for the final manuscript version in order to

- Standardize all information
- specify the doi for all references
- or including ISBN as "Persistent identifier" for book references

according to the submission guideline.

Furthermore, we have added the source (Esri) more visible within the figure 4 according to the comment from the editorial support team.